# EECE: Ensemble-based Epistemic and Cooperative Exploration for Multi-Agent Reinforcement Learning

## Abstract

Efficient exploration in multi-agent reinforcement learning (MARL) remains a fundamental challenge, particularly in complex cooperative tasks with sparse rewards. In MARL, agents must discover both novel and strongly cooperative state–action pairs in high-dimensional state–action space to effectively facilitate policy learning. In this paper, we propose Ensemble-based Epistemic and Cooperative Exploration (EECE), a unified framework that leverages an ensemble dynamics model to simultaneously capture epistemic uncertainty for directed exploration and the level of cooperation required for coordinated behavior discovery. To achieve this, EECE introduces two information-theoretic intrinsic rewards: (i) an epistemic information gain signal that directs agents toward transitions with high uncertainty, and (ii) a cooperative signal that maximizes the aggregated marginal influence of individual agents on global state variation, quantified via mutual information. It then employs a dynamic weighting strategy to leverage the complementary effects of intrinsic rewards during training. Moreover, it incorporates a dual-policy mechanism that stabilizes exploration and avoids introducing additional non-stationarity and credit assignment issues. We demonstrate the advantages of our method through cooperative benchmarks with sparse rewards, including the StarCraft Multi-Agent Challenge (SMAC) and Google Research Football (GRF), showing that EECE achieves substantial improvements in both exploration efficiency and final performance.

## 1 Introduction

Multi-Agent Reinforcement Learning (MARL) has gained considerable attention in recent years for its potential to address complex cooperative tasks involving multiple agents (Omidshafiei et al., 2017; Lowe et al., 2017). Under the Centralized Training with Decentralized Execution (CTDE) paradigm, value factorization frameworks have shown strong empirical performance across a broad set of cooperative benchmarks (Sunehag et al., 2017; Rashid et al., 2020; Wang et al., 2020). However, effective exploration remains a fundamental challenge for MARL, especially in environments with sparse rewards, which are prevalent in many real world applications (Liu et al., 2021).

In RL, effectively balancing exploration and exploitation is particularly challenging in high-dimensional environments with sparse rewards, where limited feedback hampers policy learning and increases the demand for effective exploration (Pathak et al., 2017; 2019; Sukhija et al., 2024). In MARL, achieving effective exploration presents more severe challenges (Liu et al., 2021; Zheng et al., 2021). On one hand, as the joint state-action space grows exponentially with the number of agents, single-agent methods like count-based (Tang et al., 2017) or curiosity-driven strategies (Burda et al., 2018) struggle to quantify novelty, making diverse and informative exploration particularly challenging (Zheng et al., 2021). On the other hand, the behaviors of agents are interdependent in MARL, and many tasks require cooperation to reach critical states and achieve specified goals (Wang et al., 2019; Jeon et al., 2022). Exploration should be conducted collaboratively to ensure complementary actions, while excessive non-cooperative exploration can reduce learning efficiency.

To address the challenge of exploration under sparse rewards, intrinsic rewards have become a common technique (Liu et al., 2023; Na & Moon, 2024; Jo et al., 2024). Prior approaches employ var-

ious intrinsic rewards for multi-agent settings, such as prediction-error based novelty (Zheng et al., 2021), trajectory-identity mutual information (Li et al., 2021; 2024b), influence modeling (Wang et al., 2019), or Bayesian surprise (Li et al., 2024c). Although these methods encourage either diversity or cooperation, they often fail to provide reliable exploration signals in high-dimensional state-action spaces. More importantly, they lack a unified treatment of both aspects, which limits their overall effectiveness. Therefore, developing a unified and effective exploration framework for MARL remains a significant open challenge.

Ensemble-based exploration has been proven effective in single-agent RL as a robust mechanism for generating stable exploration signals (Lee et al., 2021a; Yao et al., 2021). By leveraging prediction disagreement, ensemble methods provide a effective approach to quantify epistemic uncertainty (Lakshminarayanan et al., 2017), thereby naturally guiding exploration toward under-explored regions of the state–action space (Pathak et al., 2019; Sekar et al., 2020). Recent works have established a theoretical connection between information gain and epistemic uncertainty, which enables more reliable exploration in both model-based and model-free settings (Sukhija et al., 2023; 2024). Although ensemble methods show potential in high-dimensional environments with sparse rewards, they remain largely underexplored in MARL, where challenges such as cooperative exploration (Kim & Sung, 2023; Jo et al., 2024), credit assignment (Foerster et al., 2018), and partial observability (Hong et al., 2022; Li et al., 2024a) limit their adoption.

In this work, we introduce Ensemble-based Epistemic and Cooperative Exploration (EECE), a unified framework designed to simultaneously enhance diverse exploration and inter-agent cooperation in MARL. The core idea is to leverage an ensemble of learned dynamics models (Lakshminarayanan et al., 2017) combined with information-theoretic measures (Shannon, 1948; MacKay, 2003) to quantify the exploration value of state–action pairs, considering both novelty and the level of cooperation. Specifically, for the novelty dimension, we employ information gain as an epistemic intrinsic reward, which encourages agents to actively explore regions with high epistemic uncertainty, leading to directional exploration rather than uniform random exploration. For the cooperation dimension, mutual information is used to quantify each agent's marginal influence on global state variation. The marginal influences of all agents are then aggregated to measure the level of cooperation, serving as a cooperative intrinsic reward that encourages agents to act collaboratively and proactively. Importantly, this cooperative reward can be directly estimated using ensemble models without requiring any additional modules. These two intrinsic rewards are combined via a dynamic weighting strategy, enabling a smooth transition from diversity-driven to cooperation-oriented exploration during training and effectively producing a complementary, integrated intrinsic reward. This integrated reward is used to train independent exploration policies, which provide guidance on transition sampling, thereby encouraging agents to perform meaningful exploratory behaviors. The dual-policy mechanism effectively mitigates the additional non-stationarity and credit assignment issues introduced by intrinsic rewards, thereby stabilizing exploration. Our contributions are summarized as follows:

- A unified ensemble-based epistemic and cooperative exploration framework, EECE, is proposed to enable efficient and stable exploration in MARL.

- Novel ensemble-based intrinsic rewards are introduced, including an information gain metric for epistemic exploration and a mutual information-based cooperative reward for promoting inter-agent collaboration. Both rewards are derived from ensemble models and combined via a dynamic weighting strategy to produce an integrated intrinsic reward.

- A dual-policy mechanism is developed to enable independent learning of exploration and exploitation policies, with the exploration policies providing valuable state–action guidance to facilitate effective exploration in multi-agent scenarios.

- Extensive experiments are conducted on challenging multi-agent benchmarks, demonstrating that EECE significantly improves both exploration efficiency and final task performance compared to state-of-the-art baselines.

## 2 PRELIMINARIES

### 2.1 DECENTRALIZED POMDP

A Decentralized Partially Observable Markov Decision Process (Dec-POMDP) (Oliehoek et al., 2016) is a standard framework for cooperative multi-agent reinforcement learning. It is defined by

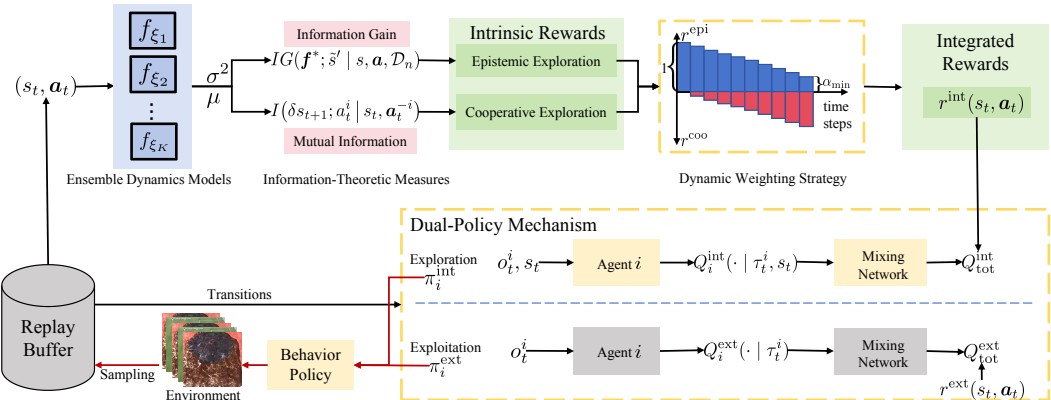

Figure 1: Overview of the EECE framework: Deep ensemble dynamics models are used with information-theoretic measures to compute epistemic and cooperative intrinsic rewards. These rewards are integrated via a dynamic weighting strategy and utilized within a dual-policy mechanism, enabling diverse exploration while fostering multi-agent cooperation.

the tuple $M = \langle \mathcal{N}, \mathcal{S}, \boldsymbol{A}, \mathcal{R}, \mathcal{P}, \boldsymbol{Z}, \boldsymbol{O}, \gamma \rangle$, where $\mathcal{N} = \{1, \ldots, n\}$ is the set of agents, $\mathcal{S}$ is the state space, and $\boldsymbol{A} = A^1 \times \cdots \times A^n$ denotes the joint action space with $A^i$ the local action space of agent $i$. At each time step $t$, agent $i$ receives a local observation $o_t^i \in Z^i$ via the observation function $O^i(o_t^i \mid s_t)$ and selects an action $a_t^i \in A^i$. The environment then transitions to the next state $s_{t+1}$ according to $\mathcal{P}(s_{t+1} \mid s_t, \boldsymbol{a}_t)$ and emits a global reward $r_t = \mathcal{R}(s_t, \boldsymbol{a}_t)$, where $\boldsymbol{a}_t = (a_t^1, \ldots, a_t^n)$ is the joint action. To handle partial observability, each agent executes a decentralized policy $\pi^i(a_t^i \mid \tau_t^i)$, where $\tau_t^i = (o_0^i, a_0^i, \ldots, o_t^i)$ is its local action-observation history. The joint policy factorizes as $\boldsymbol{\pi} = \prod_{i=1}^n \pi^i$. The objective is to learn the optimal joint policy $\boldsymbol{\pi}^* = \{\pi^{1,*}, \ldots, \pi^{n,*}\}$ that maximizes the expected discounted return: $\mathbb{E}_{\boldsymbol{\pi}, \mathcal{P}} \left[ \sum_{t=0}^{\infty} \gamma^t r_t \right]$, with discount factor $\gamma$. For training, we adopt standard value factorization methods under the CTDE paradigm, and leverage information-theoretic measures to construct intrinsic rewards, as detailed in Appendix A.

## 3 METHODOLOGY

In this section, we introduce EECE (Figure 1), a unified framework designed to enable efficient and stable exploration in multi-agent reinforcement learning.

### 3.1 DEEP ENSEMBLE DYNAMICS MODELS

EECE relies on a reliable model of environment dynamics to support epistemic and cooperative exploration. While partial observability prevents decentralized policies from accessing the global state during execution, the CTDE paradigm allows leveraging this information during training to improve model learning and exploration (Rashid et al., 2020). Therefore, we formulate the environment as a nonlinear dynamical system:

$$\tilde{s}_{t+1} = \boldsymbol{f}^*(s_t, \boldsymbol{a}_t) + w_t, \quad w_t \sim \mathcal{N}(0, \sigma^2 I), \tag{1}$$

where $\tilde{s}_{t+1} = [s_{t+1}^\top, r_t]^\top$ denotes the augmented next state including the reward, $\boldsymbol{f}^*$ represents the unknown transition and reward dynamics, and $w_t$ is zero-mean i.i.d. $\sigma^2$-Gaussian process noise. This formulation is a standard representation of nonlinear systems underlies many RL algorithms (Pathak et al., 2019; Wagenmaker et al., 2023; Sukhija et al., 2024). To approximate the unknown environment dynamics $\boldsymbol{f}^*$, an ensemble of $K$ deep neural networks is employed, providing a scalable approximation to a Bayesian dynamics model (Lakshminarayanan et al., 2017). To improve training stability and facilitate subsequent intrinsic reward computation, each network predicts $\tilde{s}_{t+1} = [\delta s_{t+1}^\top, r_t]^\top$, where $\delta s_{t+1} = s_{t+1} - s_t$ denotes the change in the global state (Figure 2). Then, given a dataset of transitions $\mathcal{D}_n = \{(s_i, \boldsymbol{a}_i, \tilde{s}_i')\}_{i=1}^n$ collected in a replay buffer, each network $f_{\xi_k}$ in the ensemble is trained independently to minimize the mean squared error (MSE) between its predictions and the targets: $f_{\xi_k} : (s_t, \boldsymbol{a}_t) \mapsto \tilde{s}_{t+1}^{(k)}, \quad \mathcal{L}(\xi_k) =$

$\frac{1}{|\mathcal{D}_n|} \sum_{(s_t,\boldsymbol{a}_t,\tilde{s}_{t+1})\in\mathcal{D}_n} \|f_{\xi_k}(s_t,\boldsymbol{a}_t) - \tilde{s}_{t+1}\|^2$. The ensemble prediction mean and variance at $(s_t,\boldsymbol{a}_t)$ are computed as

$$\mu(s_t,\boldsymbol{a}_t) = \frac{1}{K}\sum_{k=1}^{K}\tilde{s}_{t+1}^{(k)}, \quad \sigma^2(s_t,\boldsymbol{a}_t) = \frac{1}{K}\sum_{k=1}^{K}\left\|\tilde{s}_{t+1}^{(k)} - \mu(s_t,\boldsymbol{a}_t)\right\|^2, \quad (2)$$

where $\mu(s_t,\boldsymbol{a}_t) = [\mu_\delta^\top, \mu_r]^\top$ denotes the ensemble mean of state variation $\delta s_{t+1}$ and reward $r_t$, while $\sigma^2(s_t,\boldsymbol{a}_t)$ reflects the epistemic uncertainty about the unknown dynamics $\boldsymbol{f}^*$. Notably, predicting the state variation $\delta s_{t+1}$ does not change the expected variance when predicting $s_{t+1}$ and thus leaves the estimation of epistemic uncertainty unaffected. In RL, deep ensembles are effective in high-dimensional environments and robust to stochastic perturbations such as TV noise (Pathak et al., 2019). They also provide a practical approximation of the posterior distributions $p(\boldsymbol{f}^* \mid \mathcal{D}_n)$ and $p(\tilde{s}_{t+1} \mid s_t,\boldsymbol{a}_t,\mathcal{D}_n)$, which serve as the basis for computing the intrinsic rewards proposed in this work.

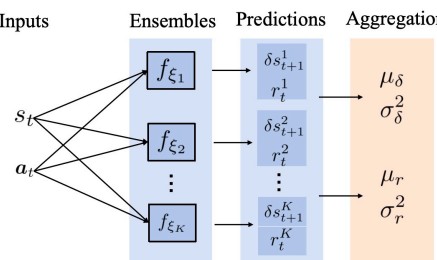

Figure 2: Deep Ensemble Dynamics Models.

### 3.2 EPISTEMIC EXPLORATION

To encourage agents to explore regions with high epistemic uncertainty, information gain is adopted as the intrinsic reward for epistemic exploration. The information gain associated with observing a transition $(s,\boldsymbol{a},\tilde{s}')$ is defined as

$$IG(\boldsymbol{f}^*; \tilde{s}' \mid s,\boldsymbol{a},\mathcal{D}_n) = H(\boldsymbol{f}^* \mid \mathcal{D}_n) - H(\boldsymbol{f}^* \mid \tilde{s}',s,\boldsymbol{a},\mathcal{D}_n), \quad (3)$$

where $H(\cdot)$ denotes Shannon differential entropy (Cover & Thomas, 2006). Theoretically, this quantity measures the reduction in uncertainty about the dynamics $\boldsymbol{f}^*$ obtained from observing a transition. Higher information gain indicates that exploring this transition can substantially improve the knowledge about the environment captured by dynamics model, thereby guiding agents toward more directed exploration. Although a posterior distribution $p(\boldsymbol{f}^* \mid \mathcal{D}_n)$ over the unknown dynamics function can be obtained using deep ensemble dynamics models, the exact computation of information gain is generally intractable. We approximate it using a tractable surrogate, as derived from epistemic uncertainty (Sukhija et al., 2023, Lemma 1). Specifically, for a transition $(s,\boldsymbol{a},\tilde{s}')$, the information gain can be upper-bounded as

$$IG\left(\boldsymbol{f}^*; \tilde{s}' \mid s,\boldsymbol{a},\mathcal{D}_n\right) \leq \underbrace{\sum_{j=1}^{d_s+1} \log\left(1 + \frac{\sigma_{n,j}^2(s,\boldsymbol{a}\mid\mathcal{D}_n)}{\sigma^2}\right)}_{r^{\mathrm{epi}}(s,\boldsymbol{a})}, \quad (4)$$

where $d_s$ denotes the dimensionality of the state space, and $\sigma^2$ corresponds to the assumed process noise. Here, $\boldsymbol{\sigma}(s,\boldsymbol{a}\mid\mathcal{D}_n) = [\sigma_j(s,\boldsymbol{a}\mid\mathcal{D}_n)]_{j=1}^{d_s+1}$ represents the per-dimension predictive variance of the ensemble, capturing epistemic uncertainty. Following Sukhija et al. (2024), we adopt the upper bound of the information gain as the intrinsic reward $r^{\mathrm{epi}}$ for epistemic exploration. Maximizing this bound intuitively guides agents toward state–action regions where the model is most uncertain about the unknown dynamics $\boldsymbol{f}^*$, thereby promoting exploration that efficiently covers state spaces.

### 3.3 COOPERATIVE EXPLORATION

In MARL, relying solely on epistemic exploration yields only limited performance gains. This limitation arises because effective exploration in multi-agent systems must occur through cooperative exploration. Without such cooperation, agents may inefficiently search the exponentially large joint action space, resulting in redundant or conflicting behaviors. Therefore, achieving efficient exploration requires combining epistemic exploration with mechanisms that explicitly encourage cooperative behavior among agents. Here, cooperation is defined as the coordinated effort of all agents to

collectively induce variations in the global state, with the actions of individual agents contributing to these changes. To encourage such cooperative behavior, we introduce a novel intrinsic reward based on the aggregated marginal influence of individual agents on global state variation, thereby promoting cooperative exploration among agents.

Motivated by information-theoretic principles, the marginal influence of individual agents on global state variation is quantified using conditional mutual information, which captures the dependency of the global state changes on individual action given the actions of other agents. Formally, for agent $i$, the conditional mutual information between its action $a_t^i$ and the resulting state variation $\delta s_{t+1}$, conditioned on the current state $s_t$ and the joint actions of all other agents $\boldsymbol{a}_t^{-i}$, is defined as

$$I\big(\delta s_{t+1}; a_t^i \mid s_t, \boldsymbol{a}_t^{-i}\big) = \mathbb{E}_{p(a_t^i \mid s_t, \boldsymbol{a}_t^{-i})}\Big[D_{\mathrm{KL}}\big(p(\delta s_{t+1} \mid s_t, a_t^i, \boldsymbol{a}_t^{-i}) \,\big\|\, p(\delta s_{t+1} \mid s_t, \boldsymbol{a}_t^{-i})\big)\Big]. \quad (5)$$

In practice, we are often interested in assessing the marginal influence of a specific action $a_t^i$ within a joint action. In this case, the conditional mutual information in equation 5 reduces to the inner KL-divergence term (Mazzaglia et al., 2022; Li et al., 2024c), which captures the causal influence of the action taken by agent $i$ on the state variation:

$$I\big(\delta s_{t+1}; a_t^i \mid s_t, \boldsymbol{a}_t^{-i}\big) = D_{\mathrm{KL}}\big(p(\delta s_{t+1} \mid s_t, a_t^i, \boldsymbol{a}_t^{-i}) \,\big\|\, p(\delta s_{t+1} \mid s_t, \boldsymbol{a}_t^{-i})\big). \quad (6)$$

The sum of these marginal influences provides a measure of the overall level of cooperation for a joint action. Accordingly, the cooperative intrinsic reward is defined as

$$r^{\mathrm{coo}}(s_t, \boldsymbol{a}_t) = \sum_{i=1}^{n} I\big(\delta s_{t+1}; a_t^i \mid s_t, \boldsymbol{a}_t^{-i}\big). \quad (7)$$

Using $r^{\mathrm{coo}}(s_t, \boldsymbol{a}_t)$ as an intrinsic reward assigns higher value to transitions where the actions of individual agents strongly influences the global state variation $\delta s_{t+1}$. This encourages agents to act coherently, promoting cooperative and efficient exploration in multi-agent settings. However, computing the conditional mutual information is generally intractable because the true conditional distributions are unknown. Fortunately, deep ensemble dynamics models provide a practical approximation of the posterior distributions, avoiding the need for additional models. By leveraging empirical predictions from the ensemble, a tractable, conservative estimate of the conditional mutual information can be obtained and used to compute the cooperative intrinsic reward. A detailed derivation and proof are provided in Appendix B.

**Proposition 1** (Ensemble-Based Empirical Estimate of Conditional Mutual Information). *Let an ensemble of $K$ learned dynamics models $f_{\xi_1}, \ldots, f_{\xi_K}$ approximate the environment transition $s_t \mapsto \delta s_{t+1}$. For agent $i$, the conditional mutual information between its action $a_t^i$ and the resulting state variation $\delta s_{t+1}$, conditioned on the other agents' actions $\boldsymbol{a}_t^{-i}$, can be empirically conservatively approximated using the ensemble statistics:*

$$I(\delta s_{t+1}; a_t^i \mid s_t, \boldsymbol{a}_t^{-i}) \approx D_{\mathrm{KL}}\Big(\mathcal{N}(\mu_\delta^i, \Sigma_\delta^i) \,\big\|\, \mathcal{N}(\mu_\delta^{-i}, \Sigma_\delta^{-i})\Big), \quad (8)$$

*where $\mu_\delta^i, \Sigma_\delta^i$ are the empirical mean and covariance of the $K$ ensemble predictions under $a_t^i$, and $\mu_\delta^{-i}, \Sigma_\delta^{-i}$ are the corresponding statistics for counterfactual predictions marginalizing out $a_t^i$.*

### 3.4 UNIFIED EXPLORATION FRAMEWORK

To leverage both epistemic and cooperative exploration, we integrate the corresponding intrinsic rewards into a unified exploration and training framework.

**Dynamic weighting strategy** Firstly, we adopt a dynamic weighting strategy that balances the contributions of epistemic and cooperative rewards over the course of training: $r^{\mathrm{int}}(s, \boldsymbol{a}) = \alpha_t \, r^{\mathrm{epi}} + (1 - \alpha_t) \, r^{\mathrm{coo}}$, where $r^{\mathrm{int}}$ is integrated intrinsic rewards, and $t$ is the cumulative number of environment interaction steps. The coefficient $\alpha_t$ evolves over training according to

$$\alpha_t = \alpha_{\min} + (1 - \alpha_{\min}) \exp\Big(-\frac{\kappa t}{T_{\max}}\Big), \quad (9)$$

where $\alpha_{\min} \in (0, 1)$ sets the final emphasis on epistemic exploration, $\kappa$ controls its decay from 1 to $\alpha_{\min}$, and $T_{\max}$ is the total training steps. The scheme shifts from early epistemic exploration to later cooperative exploration, effectively balancing the two signals for efficient multi-agent learning.

The dynamic weighting strategy is motivated by the evolving roles of exploration and cooperation during training. Early in training, agents have limited knowledge of the environment, emphasizing epistemic rewards encourages visiting novel states and reducing model uncertainty. Later, as agents acquire sufficient information, cooperative behaviors become critical for team performance. Gradually shifting the weight from epistemic to cooperative rewards allows EECE to explore effectively first and then focus on coordination, resulting in more efficient and stable multi-agent learning.

**Dual-policy mechanism for stable exploration**   Previous methods combine intrinsic $r^{\text{int}}$ and extrinsic $r^{\text{ext}}$ rewards with fixed weights, e.g., $r^{\text{tot}} = r^{\text{ext}} + \beta r^{\text{int}}$. In multi-agent settings, such naive combination worsens non-stationarity (Burda et al., 2018) and complicates credit assignment (Li et al., 2024c), destabilizing joint policy learning. To address these challenge, we propose a dual-policy mechanism. Specifically, an exploration policy $\{\pi_i^{\text{int}}\}_{i=1}^n$ is trained solely on intrinsic rewards derived from ensemble models, building on value factorization methods but leveraging privileged information during training (Hong et al., 2022). This design facilitates effective policy learning without constraining the discrete execution of the exploitation policy. In parallel, an exploitation policy $\{\pi_i^{\text{ext}}\}_{i=1}^n$ is optimized exclusively with extrinsic rewards, ensuring stable task-oriented learning. The two policies are trained in parallel and play complementary roles: the exploration policy produces novel and cooperative trajectories that enrich the experience buffer, while the exploitation policy leverages this data to enhance task performance.

In our framework, the exploration policy $\{\pi_i^{\text{int}}\}_{i=1}^n$ is used to extend the classical $\epsilon$–greedy policy into a dual-policy exploration scheme. At each environment step $t$, the joint action $\boldsymbol{a}_t = (a_t^1, a_t^2, \ldots, a_t^n)$ is sampled according to a dual-policy behavior strategy:

$$
\boldsymbol{a}_t = \begin{cases} \text{sample } \boldsymbol{a} \sim \prod_{i=1}^n \pi_i^{\text{int}}(\cdot \mid \tau_t^i, s_t), & \text{with probability } \beta, \\ \text{sample } \boldsymbol{a} \sim \prod_{i=1}^n \pi_i^{\text{ext}}(\cdot \mid \tau_t^i), & \text{otherwise}, \end{cases} \tag{10}
$$

where the policies are defined as

$$
\pi_i^{\text{int}}(\cdot \mid \tau_t^i, s_t) = \text{Softmax}\big(Q_i^{\text{int}}(\cdot \mid \tau_t^i, s_t)\big), \quad \pi_i^{\text{ext}}(\cdot \mid \tau_t^i) = \text{Greedy}\big(Q_i^{\text{ext}}(\cdot \mid \tau_t^i)\big), \tag{11}
$$

with $Q_i^{\text{int}}(\cdot \mid \tau_t^i, s_t)$ and $Q_i^{\text{ext}}(\cdot \mid \tau_t^i)$ denoting the local Q-value function learned for exploration and exploitation, respectively. Both the exploration and exploitation policies are trained by minimizing the TD-error: $\mathcal{L}(\theta) = \mathbb{E}_{\boldsymbol{\tau},\boldsymbol{a},r,\boldsymbol{\tau}'\sim\mathcal{D}}\left[\big(r + \gamma \max_{\boldsymbol{a}'} Q_{\theta^-}^{\text{tot}}(\boldsymbol{\tau}', \boldsymbol{a}') - Q_{\theta}^{\text{tot}}(\boldsymbol{\tau}, \boldsymbol{a})\big)^2\right]$, where $\theta$ and $r$ correspond to the parameters and rewards for either the exploration or exploitation policy, and $\theta^-$ denotes the parameters of the target network.

## 4 RELATED WORKS

Intrinsic rewards have been widely adopted in MARL to encourage exploration under sparse rewards. Representative methods include MAVEN (Mahajan et al., 2019), which employs hierarchical latent variables to diversify exploration. EITI and EDTI (Wang et al., 2019) maximize the mutual influence among agents' transitions and value functions. Other approaches, such as EMC (Zheng et al., 2021) and MASER (Jeon et al., 2022), enhance exploration efficiency by leveraging high-reward trajectories. Recent studies further explore information-theoretic principles. CDS (Li et al., 2021) and PMIC (Li et al., 2022) optimize mutual information to promote diversity or cooperative behaviors, while FoX (Jo et al., 2024) encourages agents to explore diverse formations. ICES (Li et al., 2024c) incorporates Bayesian surprise (Mazzaglia et al., 2022) to scaffold cooperative exploration under sparse rewards. Unlike previous approaches, EECE leverages ensemble models for stable predictions and information-theoretic measures to construct epistemic and cooperative intrinsic rewards. A dynamic weighting strategy combined with a dual-policy mechanism then integrates these rewards, guiding exploration toward informative and collaborative state-action regions in multi-agent reinforcement learning.

## 5 EXPERIMENTS

We evaluate the proposed method through a series of experiments designed to address the following key aspects: **Q1.** The performance of EECE in sparse reward multi agent settings compared to state-of-the-art MARL frameworks (Section 5.1); **Q2.** The contribution of each major component of EECE to overall performance (Section 5.2); **Q3.** The capability of EECE to discover novel states (Section 5.3); **Q4.** The emergence of cooperative behaviors under EECE (Section 5.3). We consider challenging multi-agent benchmarks, including SMAC (Samvelyan et al., 2019) and GRF (Kurach et al., 2020). For comparison, we evaluate EECE against a range of representative MARL baselines such as QMIX (Rashid et al., 2020), EMC (Zheng et al., 2021), CDS (Li et al., 2021), FOX (Jo et al., 2024), and ICES (Li et al., 2024c). We report both the mean and standard deviation of performance over five random seeds. All baseline methods are implemented with the hyperparameter configurations provided in their original works. For EECE, detailed hyperparameter settings are included in Appendix D.

### 5.1 COMPARATIVE EVALUATION ON BENCHMARK PROBLEMS

**Environmental settings** We adopt the sparse reward settings used in prior work (Kim & Sung, 2023; Li et al., 2024c). In SMAC, rewards are provided solely when allied or enemy units are eliminated, while in GRF, agents receive rewards only upon scoring or losing the game. Such sparse feedback provides limited learning signals, making effective exploration essential for success. See Appendix D.1 for environmental details.

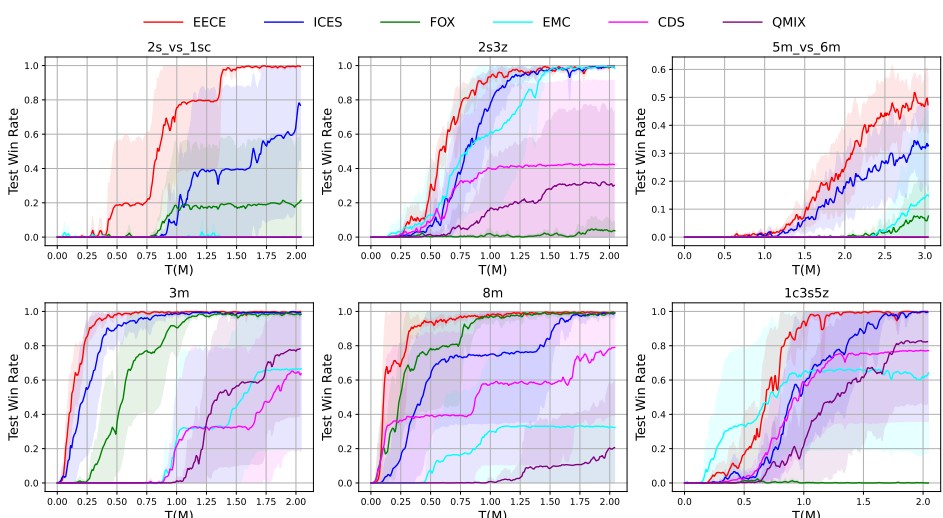

Figure 3: Performance comparison of EECE compared to baseline algorithms on SMAC task.

**SMAC** We evaluate EECE on six representative SMAC scenarios. As shown in Figure 3, leveraging ensemble-based intrinsic rewards and the dual-policy mechanism, EECE consistently outperforms state-of-the-art baselines. This shows that EECE promotes effective exploration in MARL without affecting the original training objective, yielding faster convergence and superior performance. In the `2s_vs_1sc` scenario, EECE quickly learns a strategy achieving a 100% win rate after 1.5M steps, whereas the top baseline reaches only 40% and most others stay below 20%. Although ICES achieves competitive results in some tasks by using individual contributions as intrinsic scaffolds, it lacks an explicit mechanism for promoting state-space diversity, resulting in less efficient exploration than EECE. FOX performs well in `3m` and `8m` scenarios by leveraging formation information, but its reliance on formation-level metrics without explicit action-level cooperation limits exploration in other scenarios. These results highlight EECE's key strength: by unifying epistemic and cooperative exploration with deep ensemble-based approximations of information-theoretic measures, it enables more effective exploration across diverse MARL scenarios.

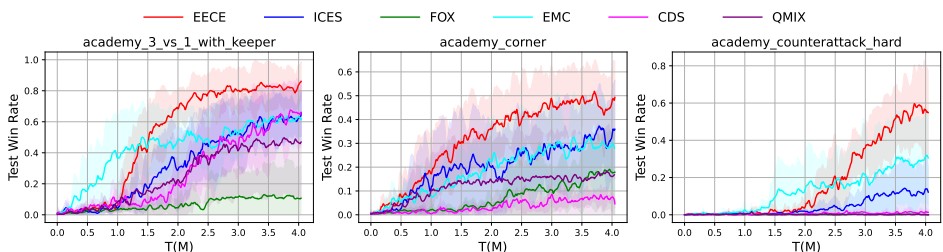

Figure 4: Performance comparison of EECE compared to baseline algorithms on GRF task.

**GRF** We evaluate EECE on three challenging GRF scenarios to assess its generality, which features richer dynamics, stochasticity, and diverse cooperative behaviors (Kurach et al., 2020). As shown in Figure 4, EECE consistently outperforms baselines. In the challenging `academy_counterattack_hard` scenario, most baselines achieve less than 30% win rate after 4M steps, with some failing completely at 0%. In contrast, EECE reaches around 60%, demonstrating its clear advantage in handling complex cooperative tasks. Although EMC with episodic control achieves higher win rates in the early stages, it often converges to suboptimal policies due to insufficient effective exploration. In contrast, EECE provides stable and informative exploration signals throughout training, enabling agents to develop stronger and more cooperative policies.

## 5.2 Ablation Studies

We perform ablations to assess the contributions of EECE's key components, including the design of intrinsic rewards and the dual-policy mechanism. We also analyze the sensitivity of EECE to key hyperparameters, namely $\alpha_{\min}$, $\kappa$, and $\beta$, demonstrating the robustness of our method. A detailed description of these experiments can be found in Appendix E.

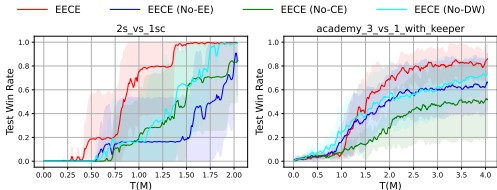 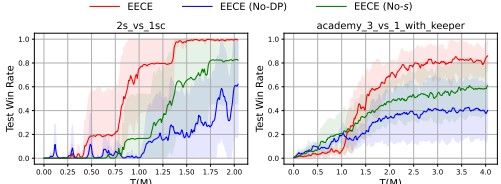

Figure 5: Ablations on intrinsic reward design.    Figure 6: Ablations on dual-policy mechanism.

**Intrinsic reward design.** We compare **EECE** with several ablated variants to evaluate the contributions of different intrinsic signals: (1) **EECE (No-EE)**, trained without the epistemic reward $r^{\mathrm{epi}}(s, \boldsymbol{a})$; (2) **EECE (No-CE)**, trained without the cooperative reward $r^{\mathrm{coo}}(s, \boldsymbol{a})$; and (3) **EECE (No-DW)**, trained without the dynamic weighting strategy, where the intrinsic reward is a fixed linear combination, $r^{\mathrm{int}} = 0.5r^{\mathrm{epi}} + 0.5r^{\mathrm{coo}}$. As shown in Figure 5, removing either reward signal leads to a significant performance drop on both SMAC and GRF tasks. Although both epistemic and cooperative exploration play critical roles across different scenarios, the degree to which each type of exploration contributes can vary depending on the specific task. Moreover, a fixed-weight combination of the two rewards underperforms compared to EECE. These results demonstrate that EECE effectively integrates both forms of intrinsic rewards, thereby achieving superior final performance.

**Dual-policy mechanism.** To evaluate the contributions of our proposed dual-policy mechanism, we conduct an ablation study comparing **EECE** with two variants: (1) **EECE (No-DP)**, where the dual-policy mechanism is disabled and the exploitation policy is trained directly with $r^{\mathrm{tot}} = r^{\mathrm{ext}} + 0.5r^{\mathrm{int}}$; and (2) **EECE (No-$s$)**, which removes access to the global state $s$ for the exploration policy, following a stricter CTDE setting. Figure 6 shows that naively adding intrinsic rewards to extrinsic signals leads to instability and substantial performance degradation. Furthermore, enforcing strict CTDE by removing global state information harms performance, as the exploration policy suffers

from larger estimation errors (Hong et al., 2022).These observations highlight that the dual-policy mechanism enables effective utilization of both intrinsic and extrinsic signals, mitigating additional non-stationarity and credit assignment issues, and thereby achieving optimal learning performance.

### 5.3 QUALITATIVE ANALYSIS

We provide qualitative results to illustrate the exploratory behaviors encouraged by EECE. For novel state discovery, we measure the diversity of visited states using SimHash-based state counting (Tang et al., 2017), and compare the number of unique states encountered by EECE and QMIX. As shown in Figure 7, EECE explores substantially more novel states, ultimately covering about 4000 regions, compared to only about 1500 with $\epsilon$-greedy in QMIX. Notably, we decay EECE's exploration rate $\beta$ from 0.1 to 0.05, whereas QMIX uses $\epsilon$-greedy with a floor of 0.1. Once $\epsilon$ reaches 0.1, QMIX rarely explores new regions, while EECE continues to expand coverage.

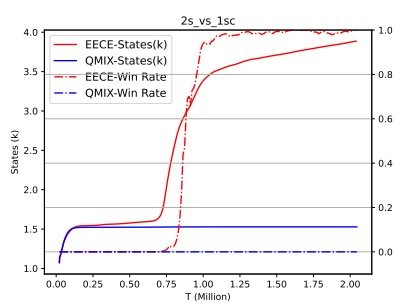

This persistent exploration enables EECE to discover critical state-actions at around 750k steps, leading to rapid policy improvement and a substantial win-rate increase. These results demonstrate EECE's ability to discover cooperative novel states. Details of the SimHash-based state counting and configuration are provided in Appendix F.

To illustrate cooperative behaviors, we visualize representative actions guided by the policies of EECE. As shown in Figure 8, the two Stalkers approach the Spine Crawler from different directions, coordinating their attacks while one unit draws enemy fire, resulting in a successful joint elimination. In Figure 9, EECE enables teammates to pass and shoot effectively, ultimately winning the match. These observations indicate that EECE not only promotes diverse state exploration but also fosters cooperative behaviors that accelerate learning.

Figure 7: Number of visited states and win rate of EECE and QMIX on the 2s_vs_1sc.

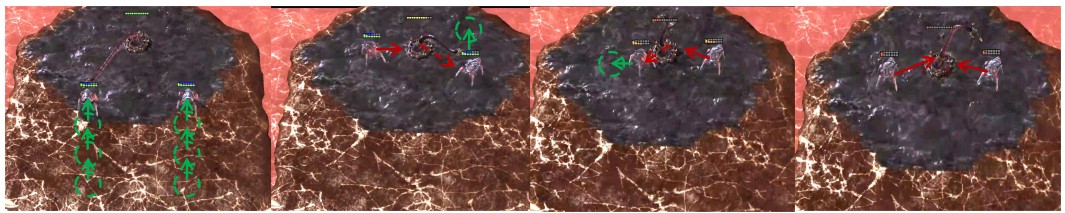

Figure 8: Visualization of exploration policy induced by EECE on the 2s_vs_1sc scenario. Solid arrows indicate greedy actions selected by the exploration policy. Green arrows denote movement, red arrows denote attack, and dashed red arrows indicate enemy attack (Appendix G).

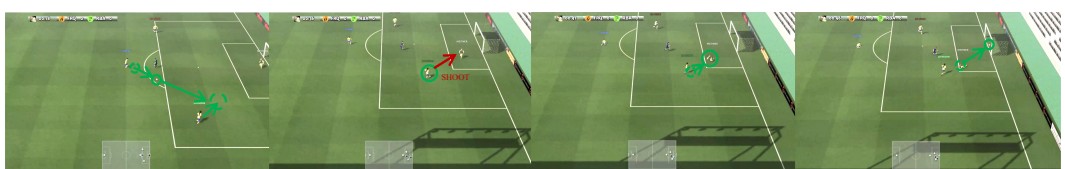

Figure 9: Visualization of exploitation policy in the academy_3_vs_1_with_keeper scenario. Green arrows indicate the movements of ball or teammates, while red arrows represent shooting.

## 6 CONCLUSIONS

In this work, we introduced EECE, a unified multi-agent exploration framework that promotes both epistemic and cooperative exploration. It leverages deep ensemble dynamics models combined with

information-theoretic measures to generate stable and effective intrinsic rewards, which are integrated via a dynamic weighting strategy and learned through a dual-policy mechanism that decouples exploration from exploitation. Experiments demonstrate that EECE consistently outperforms state-of-the-art baselines across diverse SMAC and GRF benchmarks.

**Limitations and future works.** EECE requires training an ensemble of forward dynamics models to compute intrinsic rewards, as well as an exploration policy, which adds computational overhead. In addition, in realistic settings, effective cooperation is often task-driven, and exploring task-aware cooperative rewards is a promising direction, for example via goal-conditioned RL (Nasiriany et al., 2019; Na & Moon, 2024) or episodic control (Pritzel et al., 2017; Lin et al., 2018) techniques. Developing adaptive mechanisms to combine these two types of exploration rewards is also an interesting avenue for future research.

## REPRODUCIBILITY STATEMENT

Pseudocode is provided in Appendix C. For the theoretical aspects, detailed proofs are included in Appendix B. For the practical aspects, the experimental setup is described in Section 5, and hyperparameters and implementation details are provided in Appendix D. The code is included in the supplementary material.

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

## A  ADDITIONAL PRELIMINARIES

### A.1  CENTRALIZED TRAINING WITH DECENTRALIZED EXECUTION (CTDE)

CTDE is a promising paradigm in deep cooperative multi-agent reinforcement learning, where the local agents execute actions only based on local observation histories, while the policies can be trained in centralized manner which has access to global information (Lowe et al., 2017; Yu et al., 2022; Sunehag et al., 2017). Under Centralized Training with Decentralized Execution (CTDE) framework, value factorization approaches (Sunehag et al., 2017; Rashid et al., 2020; Wang et al., 2020) have been introduced to solve fully cooperative multi-agent reinforcement learning (MARL) tasks, and these approaches achieved state-of-the-art performance in challenging benchmark problems such as SMAC (Samvelyan et al., 2019). Value factorization approaches utilize the joint action-value function $Q_\theta^{\text{tot}}$ with learnable parameter $\theta$. Then, the training objective $\mathcal{L}(\theta)$ can be expressed as

$$\mathcal{L}(\theta) = \mathbb{E}_{\boldsymbol{\tau}, \boldsymbol{a}, r, \boldsymbol{\tau}' \sim \mathcal{D}} \left[ \left( r + \gamma \max_{\boldsymbol{a}'} Q_{\theta^-}^{\text{tot}} \left( \boldsymbol{\tau}', \boldsymbol{a}' \right) - Q_\theta^{\text{tot}} \left( \boldsymbol{\tau}, \boldsymbol{a} \right) \right)^2 \right], \tag{12}$$

where $\mathcal{D}$ is the replay buffer and $\theta^-$ denotes the parameters of the target network, which is periodically updated by $\theta$. Each local agent conducts decision-making using its individual utility $Q_i(\tau^i, a^i)$, which acts as a proxy for its Q-function.

### A.2  INFORMATION-THEORETIC MEASURES

Information-theoretic measures (Shannon, 1948; Cover & Thomas, 2006; MacKay, 2003) provide principled ways to quantify uncertainty and dependencies between random variables. Two commonly used measures are information gain and mutual information.

**Information Gain.** Information gain quantifies the reduction in uncertainty about a random variable $X$ after observing another variable $Y$:

$$IG(X; Y) = H(X) - H(X \mid Y), \tag{13}$$

where $H(X)$ is the entropy of $X$ and $H(X \mid Y)$ is the conditional entropy of $X$ given $Y$. Information gain measures how much knowledge of $Y$ reduces uncertainty in $X$.

**Mutual Information.** Mutual information quantifies the dependency between two random variables $X$ and $Y$:

$$I(X;Y) = \mathbb{E}_Y\Big[D_{\mathrm{KL}}\big(p(X \mid Y) \,\|\, p(X)\big)\Big], \tag{14}$$

where $D_{\mathrm{KL}}(p \,\|\, q) = \mathbb{E}_p\big[\log \frac{p}{q}\big]$ denotes the Kullback–Leibler (KL) divergence between distributions $p$ and $q$, which measures the difference between them. Unlike information gain, which is directional, mutual information is symmetric and captures the overall amount of shared information between $X$ and $Y$.

## B  APPROXIMATING MUTUAL INFORMATION

**Proposition 1** (Ensemble-Based Empirical Estimate of Conditional Mutual Information). *Let an ensemble of $K$ learned dynamics models $f_{\xi_1}, \ldots, f_{\xi_K}$ approximate the environment transition $s_t \mapsto \delta s_{t+1}$. For agent $i$, the conditional mutual information between its action $a_t^i$ and the resulting state variation $\delta s_{t+1}$, conditioned on the other agents' actions $\boldsymbol{a}_t^{-i}$, can be empirically approximated using the ensemble statistics:*

$$I(\delta s_{t+1}; a_t^i \mid s_t, \boldsymbol{a}_t^{-i}) \approx D_{\mathrm{KL}}\Big(\mathcal{N}(\mu_\delta^i, \Sigma_\delta^i) \,\Big\|\, \mathcal{N}(\mu_\delta^{-i}, \Sigma_\delta^{-i})\Big), \tag{15}$$

*where $\mu_\delta^i, \Sigma_\delta^i$ are the empirical mean and covariance of the $K$ ensemble predictions under $a_t^i$, and $\mu_\delta^{-i}, \Sigma_\delta^{-i}$ are the corresponding statistics for counterfactual predictions marginalizing out $a_t^i$.*

*Proof.* For a specific action $a_t^i$ in a given joint action, the conditional mutual information simplifies to the inner KL-divergence term:

$$I\big(\delta s_{t+1}; a_t^i \,\big|\, s_t, \boldsymbol{a}_t^{-i}\big) = D_{\mathrm{KL}}\Big(p(\delta s_{t+1} \mid s_t, a_t^i, \boldsymbol{a}_t^{-i}) \,\Big\|\, p(\delta s_{t+1} \mid s_t, \boldsymbol{a}_t^{-i})\Big). \tag{16}$$

Let an ensemble of $K$ learned dynamics models produce $K$ vector predictions:

$$\delta s_{t+1}^{(1)}, \ldots, \delta s_{t+1}^{(K)} \in \mathbb{R}^d$$

for the same input $(s_t, a_t^i, \boldsymbol{a}_t^{-i})$. We construct an empirical approximation of the conditional distribution as a Gaussian with mean and covariance computed from the ensemble:

$$\hat{p}(\delta s_{t+1} \mid s_t, a_t^i, \boldsymbol{a}_t^{-i}) = \mathcal{N}(\mu_\delta^i, \Sigma_\delta^i), \tag{17}$$

where

$$\mu_\delta^i = \frac{1}{K}\sum_{k=1}^K \delta s_{t+1}^{(k)}, \quad \Sigma_\delta^i = \frac{1}{K}\sum_{k=1}^K (\delta s_{t+1}^{(k)} - \mu_\delta^i)(\delta s_{t+1}^{(k)} - \mu_\delta^i)^\top. \tag{18}$$

Similarly, the counterfactual marginal distribution over $a_t^i$ is approximated empirically as

$$\hat{p}(\delta s_{t+1} \mid s_t, \boldsymbol{a}_t^{-i}) = \mathcal{N}(\mu_\delta^{-i}, \Sigma_\delta^{-i}), \tag{19}$$

where the mean and covariance are computed by averaging over both the ensemble members and all possible actions of agent $i$:

$$\mu_\delta^{-i} = \frac{1}{K|A^i|}\sum_{a_t^i \in A^i}\sum_{k=1}^K \delta s_{t+1}^{(k, a_t^i)}, \quad \Sigma_\delta^{-i} = \frac{1}{K|A^i|}\sum_{a_t^i \in A^i}\sum_{k=1}^K (\delta s_{t+1}^{(k, a_t^i)} - \mu_\delta^{-i})(\delta s_{t+1}^{(k, a_t^i)} - \mu_\delta^{-i})^\top. \tag{20}$$

These empirical Gaussian distributions preserve the first- and second-order statistics of the ensemble predictions, while ignoring higher-order moments and dependencies. Following the maximum entropy principle, a Gaussian distribution with the same mean and covariance as the ensemble predictions has maximal entropy (Cover & Thomas, 2006). As a result, $D_{\mathrm{KL}}(\hat{p}(\delta s_{t+1} \mid s_t, a_t^i, \boldsymbol{a}_t^{-i}) \,\big\|\, \hat{p}(\delta s_{t+1} \mid s_t, \boldsymbol{a}_t^{-i}))$ typically underestimates the true KL divergence, making it a conservative empirical estimate of the conditional mutual information:

$$I(\delta s_{t+1}; a_t^i \mid s_t, \boldsymbol{a}_t^{-i}) \approx D_{\mathrm{KL}}\Big(\mathcal{N}(\mu_\delta^i, \Sigma_\delta^i) \,\Big\|\, \mathcal{N}(\mu_\delta^{-i}, \Sigma_\delta^{-i})\Big). \tag{21}$$

In practice, this provides a tractable estimate suitable for computing cooperative intrinsic rewards. $\qquad\square$

# C  Overall Learning Framework for EECE

---

**Algorithm 1** EECE: Ensemble-based Epistemic and Cooperative Exploration

---

**Require:** Environment $\mathcal{E}$, ensemble size $K$, agents $i = 1, \ldots, n$, intrinsic reward weights $\alpha_t$, exploration probability $\beta$, batch size $B$
1: Initialize ensemble of forward dynamics models $\{f_{\xi_k}\}_{k=1}^K$
2: Initialize exploration policies $\{\pi_i^{\text{int}}\}_{i=1}^n$ and exploitation policies $\{\pi_i^{\text{ext}}\}_{i=1}^n$
3: Initialize replay buffer $\mathcal{D}$
4: **for** $t = 1$ to $T_{\max}$ **do**
5:     Interact with the environment using the behavior policy (Eq. 10) to obtain a trajectory $\mathcal{T}$
6:     Append $\mathcal{T}$ to the replay buffer $\mathcal{D}$
7:     Sample $B$ trajectories $[\mathcal{T}]_{i=1}^B \sim \mathcal{D}$
8:     Update dynamic weighting coefficient $\alpha_t$ according to the schedule
9:     For the sampled batch $[\mathcal{T}]_{i=1}^B$, compute the integrated intrinsic reward (using Eqs. 4 and 7):

$$r_t^{\text{int}} = \alpha_t r_t^{\text{epi}} + (1 - \alpha_t) r_t^{\text{coo}}$$

10:     Update exploitation policies $\{\pi_i^{\text{ext}}\}_{i=1}^n$ using the extrinsic reward $r_t^{\text{ext}}$
11:     Update exploration policies $\{\pi_i^{\text{int}}\}_{i=1}^n$ using the integrated intrinsic reward $r_t^{\text{int}}$
12:     Update ensemble models $\{f_{\xi_k}\}_{k=1}^K$ using samples from $\mathcal{D}$
13: **end for**

---

# D  Experiment details

## D.1  Environmental Settings

In this section, we describe the environments used in our experiments, namely SMAC (Samvelyan et al., 2019) and GRF (Kurach et al., 2020). To ensure effective and fair evaluation of EECE, we follow the sparse reward settings adopted in prior work (Kim & Sung, 2023; Li et al., 2024c).

**StarCraft Multi-agent Challenge (SMAC).**  In SMAC, agents are divided into two teams and must cooperate with allies while competing against enemy units controlled by the built-in game AI. At each timestep, each agent selects an action from a discrete action space, including *no-op*, *move [direction]*, *attack [enemy id]*, and *stop*. Through these actions, agents navigate and fight in continuous spatial maps. To evaluate the effectiveness and generality of our approach, we conduct experiments on six representative scenarios: 2s_vs_1sc, 3m, 8m, 2s3z, 1c3s5z, and 5m_vs_6m, as specified in Table 1. The rewards are only given upon the death of units (allies or enemies), and details are listed in Table 2. We note that performance comparisons are only meaningful within the same SMAC version due to environment updates. All experiments in this paper are conducted on SMAC version SC2.4.10 for consistency.

Table 1: StarCraft Multi-Agent Challenge (SMAC) scenarios.

| Map Name | Ally Units | Enemy Units | Scenario Type |
|---|---|---|---|
| 2s_vs_1sc | 2 Stalkers | 1 Spine Crawler | Micro-trick: alternating fire |
| 3m | 3 Marines | 3 Marines | Homogeneous & symmetric |
| 8m | 8 Marines | 8 Marines | Homogeneous & symmetric |
| 2s3z | 2 Stalkers & 3 Zealots | 2 Stalkers & 3 Zealots | Heterogeneous & symmetric |
| 1c3s5z | 1 Colossus, 3 Stalkers & 5 Zealots | 1 Colossus, 3 Stalkers & 5 Zealots | Heterogeneous & symmetric |
| 5m_vs_6m | 5 Marines | 6 Marines | Homogeneous & asymmetric |

**Google Research Football (GRF).**  GRF (Kurach et al., 2020) is a physics-based football simulator that has been widely used to evaluate cooperative multi-agent reinforcement learning algorithms. At each timestep, agents select from a discrete set of high-level actions, such as *move [direction]*, *pass*, *shoot*, while the low-level control and ball dynamics are handled by the environment. The game is played in continuous two-dimensional fields, where agents must coordinate to advance the

Table 2: StarCraft Multi-Agent Challenge (SMAC) rewards.

| Event | Reward |
|---|---|
| All enemies die | +200 |
| One enemy dies | +10 |
| One ally dies | -5 |

ball, defend, and score goals. To ensure fair evaluation under sparse reward conditions, we adopt the same settings as prior work (Kim & Sung, 2023; Li et al., 2024c), where agents only receive a reward signal when scoring or losing the game. This sparse reward structure requires strong cooperation among agents and is further complicated by the stochastic behaviors of opponents. We evaluate EECE on several representative GRF tasks, including `academy_3_vs_1_with_keeper`, `academy_corner` and `academy_counterattack_hard`. The detailed reward settings for each task are summarized in Table 3.

Table 3: Google Research Football (GRF) rewards.

| Event | Reward |
|---|---|
| Our team scores | +100 |
| Opponent team scores | -1 |
| Our team or the ball returns to our half-court | -1 |

## D.2 Implementation Details

The proposed EECE framework consists of two main modules. First, deep ensemble dynamics models serve as the foundation for intrinsic rewards, which are computed using information-theoretic measures. Second, a dual-policy mechanism leverages these intrinsic rewards to facilitate efficient exploration. In our experiments, all hyperparameters except those newly introduced are kept unchanged.

**Deep Ensemble Dynamics Models.** For training the deep ensemble dynamics models, we use fully connected neural networks as individual predictors. Each network takes the current state $s_t$ and the joint action $\boldsymbol{a}_t$ as input and predicts the state change $\delta s_{t+1} = s_{t+1} - s_t$ as well as the reward $r_t$. By including $s_t$ as part of the input, the dynamics models learn the state delta $\delta s_{t+1}$, which is analogous to the residual learning approach in deep residual networks (He et al., 2016) and promotes more stable training. Predicting $\delta s_{t+1}$ instead of $s_{t+1}$ does not change the variance of predictions, and thus preserves the epistemic uncertainty captured by the ensemble. In our experiments, the training samples $\mathcal{D}_n$ for the ensemble dynamics models are drawn from the same distribution as the experiences for policy learning. At each training step, the discrete joint action $\boldsymbol{a}_t$ is first encoded into an embedding and then concatenated with $s_t$ before being fed into the models. We optimize the model parameters using the Adam optimizer. The hyperparameters of the ensemble models are summarized in Table 4.

Table 4: Hyperparameters of the Deep Ensemble Dynamics Models.

| Environment | Features | Num Heads ($K$) | Action Embedding Dim | Learning Rate | Optimizer |
|---|---|---|---|---|---|
| SMAC | (256, 256) | 5 | 4 | 0.001 | Adam |
| GRF | (128, 128) | 5 | 4 | 0.001 | Adam |

**Intrinsic Rewards** For intrinsic rewards, the epistemic intrinsic reward $r^{\mathrm{epi}}$ is approximated based on the epistemic uncertainty $\sigma^2(s, \boldsymbol{a})$, while the cooperative intrinsic reward $r^{\mathrm{coo}}$ is computed using the empirical distribution $p(\delta s_{t+1} \mid \cdot)$. Note that the empirical distribution $p(\delta s_{t+1} \mid s_t, \boldsymbol{a}_t^{-i})$

cannot be directly obtained from the ensemble models. Inspired by counterfactual baselines (Foerster et al., 2018), we estimate the counterfactual marginal distribution $\hat{p}(\delta s_{t+1} \mid s_t, \boldsymbol{a}_t^{-i})$ using the ensemble predictions. Specifically, for each alternative action of agent $i$, we replace $a_t^i$ in the joint action and collect the predicted $\delta s_{t+1}$ from the ensemble. Aggregating these predictions across all possible actions provides an approximation of the marginal distribution. Each action's marginal impact is then normalized so that their sum reflects the overall level of cooperation. Before being combined via the dynamic weighting strategy, both $r^{\text{epi}}$ and $r^{\text{coo}}$ are appropriately standardized. The minimum weighting factor $\alpha_{\min}$ is generally set to $0.4$ or $0.2$ to ensure sustained exploration of the environment. The hyperparameters of the dynamic weighting strategy are summarized in Table 5.

Table 5: Hyperparameters of the Dynamic Weighting Strategy.

| Environment | Scenario | $\alpha_{\min}$ | $\kappa$ |
|---|---|---|---|
| SMAC | 2s_vs_1sc | 0.4 | 4 |
| | 3m | 0.4 | 4 |
| | 8m | 0.2 | 4 |
| | 2s3z | 0.2 | 4 |
| | 1c3s5z | 0.2 | 4 |
| | 5m_vs_6m | 0.4 | 1 |
| GRF | all scenarios | 0.4 | 4 |

**Dual-Policy Mechanism.** In the dual-policy mechanism, we adopt the standard QMIX to construct the exploitation policy, which strictly follows the CTDE (Centralized Training with Decentralized Execution) framework (Rashid et al., 2020), allowing discrete execution during testing. For the exploration policy, however, we relax the strict CTDE constraint by incorporating the global state $s$ into the policy input, which facilitates more effective learning (Hong et al., 2022). Moreover, to reduce computational overhead and provide reliable exploration guidance rapidly, we employ VDN as the mixing function for the exploration policy. During environment interactions, the behavioral policy is a probabilistic mixture of the exploitation and exploration policies. Specifically, the exploration probability $\beta$ is linearly decayed to a minimum value $\beta_{\min}$ to balance exploration and exploitation. The hyperparameters of the dual-policy mechanism are summarized in Table 6.

Table 6: Hyperparameters of the Dual-Policy Mechanism.

| Environment | Scenario | $\beta$ | $\beta_{\min}$ |
|---|---|---|---|
| SMAC | all scenarios | 0.1 | 0.05 |
| GRF | all scenarios | 0.2 | 0.05 |

### D.3 Infrastructure and Code Implementation

For our experiments, we mainly use GeForce RTX 3090 GPUs. Our implementation builds upon PyMARL (Samvelyan et al., 2019), PyMARL 2 (Hu et al., 2021), and the open-sourced code from ICES (Li et al., 2024c). Following previous works, we adopt PyMARL for GRF and PyMARL 2 for SMAC.

## E Additional Experimental Results

### E.1 Effect of $\beta$

In the dual-policy mechanism, $\beta$ controls the probability of selecting the exploration policy during sampling, thereby serving as a critical parameter for balancing exploration and exploitation. During training, this probability is linearly annealed from $\beta$ to a minimum value of $\beta_{\min} = 0.05$, ensuring that the agent gradually shifts from exploration to exploitation while still maintaining a non-zero chance of exploration in the later stages. This design prevents premature convergence to suboptimal strategies and guarantees sufficient policy refinement. Figure 10 illustrates the performance of

EECE under different values of $\beta$ on both SMAC and GRF tasks. From the results, we observe that EECE achieves competitive final performance across all tested values of $\beta$, which indicates the robustness of the framework. Nevertheless, the optimal choice of $\beta$ exhibits scenario dependency. In practice, selecting $\beta$ according to the characteristics of the task can lead to improved efficiency and performance.

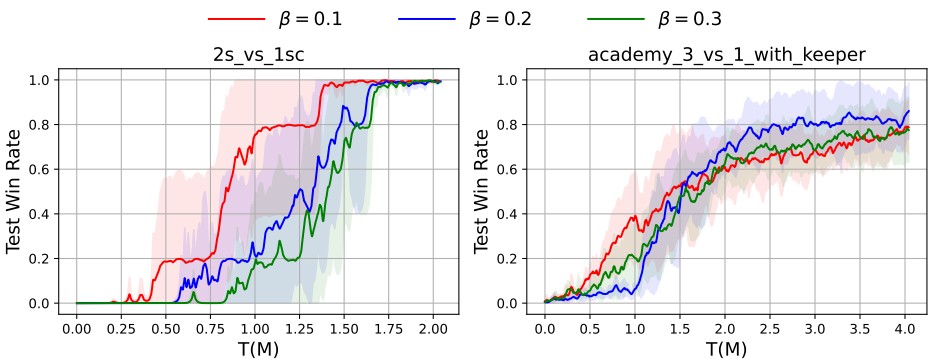

Figure 10: Ablations on $\beta$ in the `2s_vs_1sc` and `academy_3_vs_1_with_keeper` scenarios.

### E.2 EFFECT OF $\alpha_{\mathrm{MIN}}$ AND $\kappa$

$\alpha_{\min}$ and $\kappa$ are the key hyperparameters of the dynamic weighting strategy, which determine how epistemic exploration decays over training. Recall that the integrated intrinsic reward is defined as

$$r^{\mathrm{int}}(s, \boldsymbol{a}) = \alpha_t\, r^{\mathrm{epi}} + (1 - \alpha_t)\, r^{\mathrm{coo}}, \tag{22}$$

where $r^{\mathrm{epi}}$ denotes epistemic exploration rewards and $r^{\mathrm{coo}}$ denotes cooperative rewards. The weighting coefficient $\alpha_t$ evolves with training steps $t$ according to

$$\alpha_t = \alpha_{\min} + (1 - \alpha_{\min}) \exp\left( -\frac{\kappa\, t}{T_{\max}} \right), \tag{23}$$

where $T_{\max}$ is the maximum number of environment steps. Intuitively, $\alpha_t$ starts to 1, prioritizing epistemic exploration in the early stage, and gradually decays toward $\alpha_{\min}$, thereby shifting the focus to cooperative exploration as training progresses. Figure 11 illustrates how different combinations of $\alpha_{\min}$ and $\kappa$ shape the trajectory of $\alpha_t$ over time. We analyze the sensitivity of EECE to these hyper-

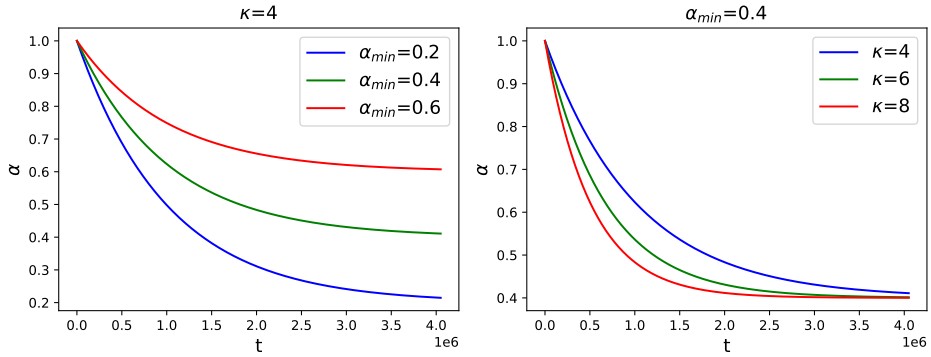

Figure 11: Evolution of $\alpha_t$ under different settings of $\alpha_{\min}$ and $\kappa$.

parameters. Figure 12 and 13 reports the ablation results in the `academy_3_vs_1_with_keeper` and `2s_vs_1sc` scenario. We observe that all tested parameter settings ultimately achieve competitive win rates, which further demonstrates the robustness of our method. In GRF, smaller values

of $\alpha_{\min}$ and larger values of $\kappa$ consistently lead to improved performance. This indicates that in this environment, cooperative exploration plays a more critical role than pure epistemic exploration. In other words, although early-stage epistemic bonuses facilitate rapid discovery of novel states, sustained emphasis on cooperative behavior is essential for solving tasks that require coordinated strategies. This observation aligns with the characteristics of the GRF benchmark, where success strongly depends on agents' ability to learn role-specialized and synergistic policies. In contrast, for SMAC tasks, a larger $\alpha_{\min}$ and a smaller $\kappa$ lead to better performance, suggesting that epistemic exploration plays a more dominant role. Therefore, carefully tuning these hyperparameters to the characteristics of each benchmark can further enhance overall performance.

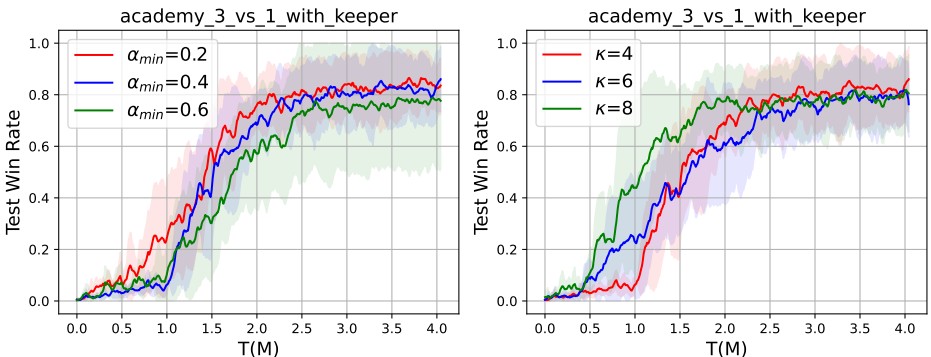

Figure 12: Performance of EECE under different $\alpha_{\min}$ and $\kappa$ settings in the `academy_3_vs_1_with_keeper` scenario.

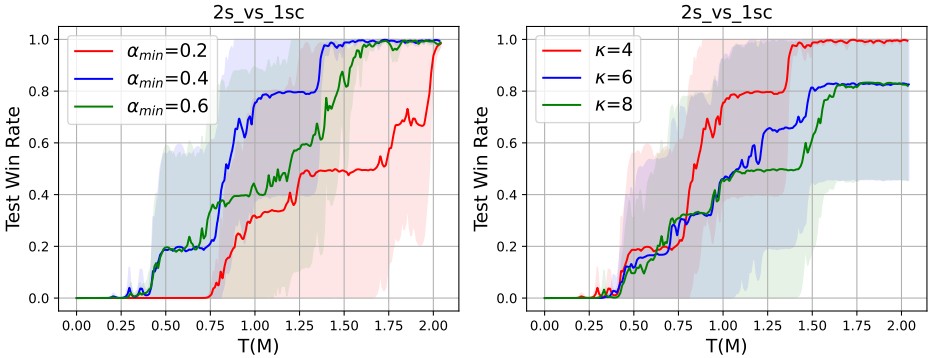

Figure 13: Performance of EECE under different $\alpha_{\min}$ and $\kappa$ settings in the `2s_vs_1sc` scenario.

## F  SIMHASH-BASED STATE COUNTING

To evaluate EECE's ability to discover novel states, we measure the diversity of visited states using SimHash-based state counting (Tang et al., 2017), and compare the number of unique states encountered by EECE and QMIX. Let $s \in \mathcal{S}$ denote a high-dimensional continuous state. Directly counting unique states in such a space is intractable due to the curse of dimensionality. To address this, we adopt SimHash, which projects continuous states into a compact discrete representation while approximately preserving similarity.

Formally, given a state $s$, SimHash projects $s$ into a $k$-bit binary code by applying a randomly initialized projection matrix $\boldsymbol{B} \in \mathbb{R}^{k \times D}$:

$$\phi(s) = \text{sign}(\boldsymbol{B}g(s)) = [\mathbb{I}(b_1 g(s) \geq 0), \dots, \mathbb{I}(b_k g(s) \geq 0)], \tag{24}$$

where $g : \mathcal{S} \rightarrow \mathbb{R}^D$ is an optional preprocessing function and $b_i$ is the $i$-th row of $\boldsymbol{B}$, sampled from a standard Gaussian distribution, $\mathbb{I}(\cdot)$ is the indicator function. The value for $k$ controls the granularity: higher values lead to fewer collisions and are thus more likely to distinguish states.

This transformation effectively discretizes the continuous state space into $2^k$ distinct partitions, allowing transitions originating from perceptually similar regions to be grouped together efficiently. By maintaining a count of these discrete codes during training, we can estimate the diversity of visited states and quantify the exploration behavior of EECE relative to baseline algorithms. In our experiments, we set $k = 16$, which provides a good trade-off between state resolution and computational efficiency. Figure 14 compares the number of visited states and the win rates of EECE and QMIX on the `academy_3_vs_1_with_keeper` task. For fairness, we set $\beta_{\min}$ in EECE and $\epsilon_{\min}$ in QMIX both to $0.05$. As shown in Figure 14, EECE visits substantially more state regions within the same number of timesteps. Notably, after around $1.25M$–$1.50M$ steps, EECE discovers critical state–action pairs that rapidly boost the win rate to $80\%$, whereas QMIX still fails to learn how to score. This clearly demonstrates EECE's ability to uncover novel states.

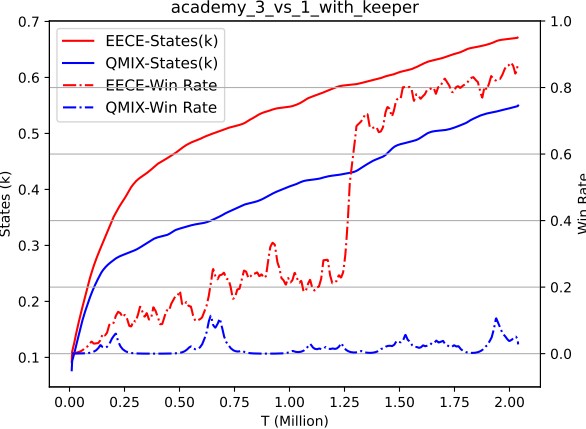

Figure 14: Number of visited states and win rate of EECE and QMIX on the `academy_3_vs_1_with_keeper`.

## G   VISUALIZATION OF POLICY

To validate the effectiveness of our framework, we visualize the exploration policy learned by EECE in the `2s_vs_1sc` scenario. In this environment, two Stalker units must cooperate to eliminate a single Spine Crawler. The main challenge lies in executing the *alternating fire* strategy: the Stalkers must take turns drawing enemy fire while the other attacks, requiring precise coordination for successful elimination. Figures 15(a) and (b) show the actions of both Stalkers under the exploration policy. The agents approach the Spine Crawler from different directions, coordinating their movements and attacks. Specifically, one Stalker temporarily draws the Spine Crawler's attention while the other launches attacks from a safer angle, resulting in a successful joint elimination. This visualization demonstrates that EECE encourages not only diverse state visitation but also emergent cooperative behaviors. Importantly, such cooperative actions (jointly moving toward the Spine Crawler) emerge even early in an episode, indicating that the exploration policy effectively captures long-term exploration value while promoting teamwork.

In the GRF environment, at each timestep, agents select from a discrete set of high-level actions, such as moving in a given direction, passing, or shooting, while low-level control and ball dynamics are managed by the environment. This scenario imposes a higher requirement for cooperation: teammates must learn to pass the ball to the player closest to the goal before attempting a shot in order to secure a win. To verify that EECE has learned such high-level cooperative strategies, we visualize a trajectory sampled using the policy learned by EECE in the `academy_3_vs_1_with_keeper` scenario. Figures 16(a) and (b) show different stages of the episode. The visualization illustrates that the agents coordinate their movements and passes effectively, positioning themselves to create scoring opportunities. For instance, a teammate strategically passes the ball to another agent closer to the goal, enabling a successful shot and demonstrating that the policy effectively captures emergent high-level cooperative behaviors.

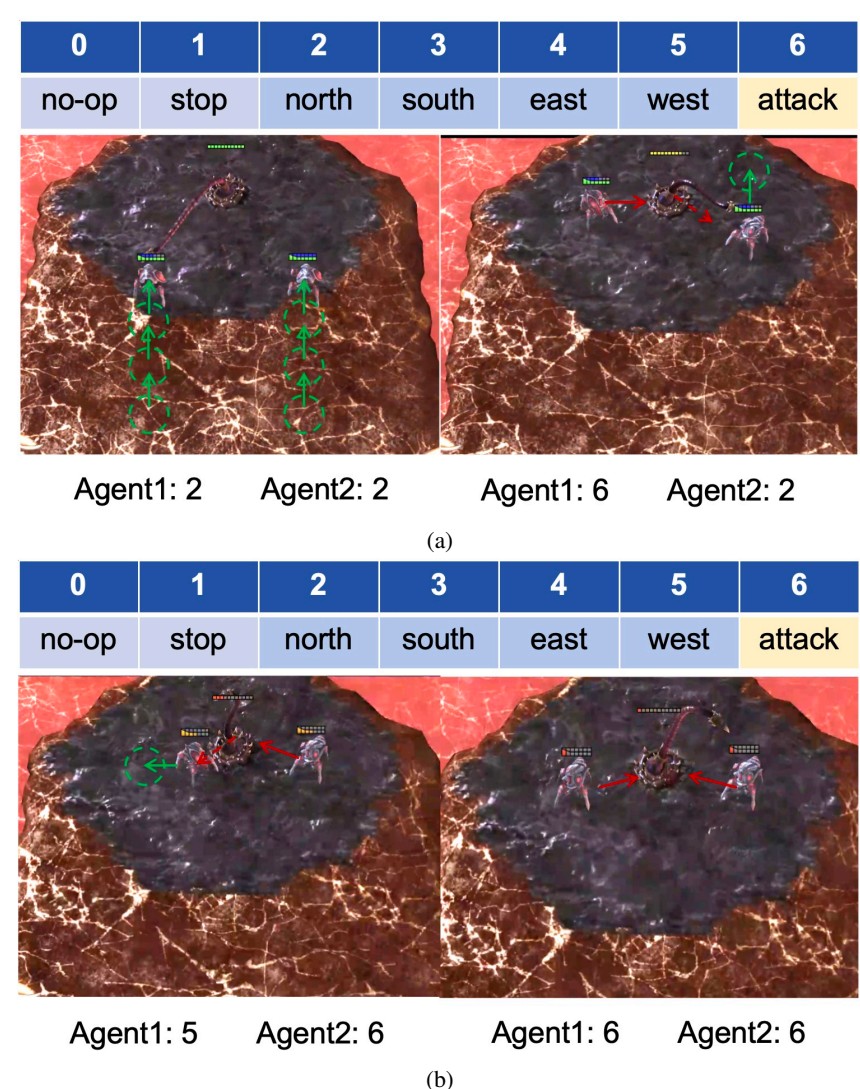

Figure 15: Visualization of the exploration policy induced by EECE in the `2s_vs_1sc` scenario. Solid arrows indicate greedy actions selected by the exploration policy. Green arrows represent movement, red arrows indicate attack, and dashed red arrows denote enemy attacks. Both subfigures illustrate how the Stalkers coordinate their positions and actions to alternate drawing enemy fire and attacking, highlighting emergent cooperative behavior facilitated by EECE in an episode.

## H  TRAINING TIME.

Compared to standard baselines, EECE requires additional training of an ensemble of forward dynamics models as well as the exploration policy, which inevitably increases computational overhead. To evaluate the training efficiency of EECE, we report the average training time per scenario and compare it against existing baselines. It is worth noting that PyMARL and PyMARL 2 differ in whether training is executed with an *episode runner* or a *parallel runner*, which has a significant impact on the total training time. The training modes,batch_size and average training times are summarized in Tables 7, 8 and 9, respectively. In Table 9, we can see that training of EECE does not take much time compared to existing baseline algorithms on SMAC tasks. In GRF, EMC requires the longest training time, whereas EECE increases the training duration by only a few hours compared to CDS, FOX, and ICES. The difference is primarily due to CDS, FOX, and ICES leveraging parallel computation for agent action selection, whereas EECE does not employ such parallelization in GRF. However, in SMAC, the training time of EECE is comparable to that of ICES, indicating that

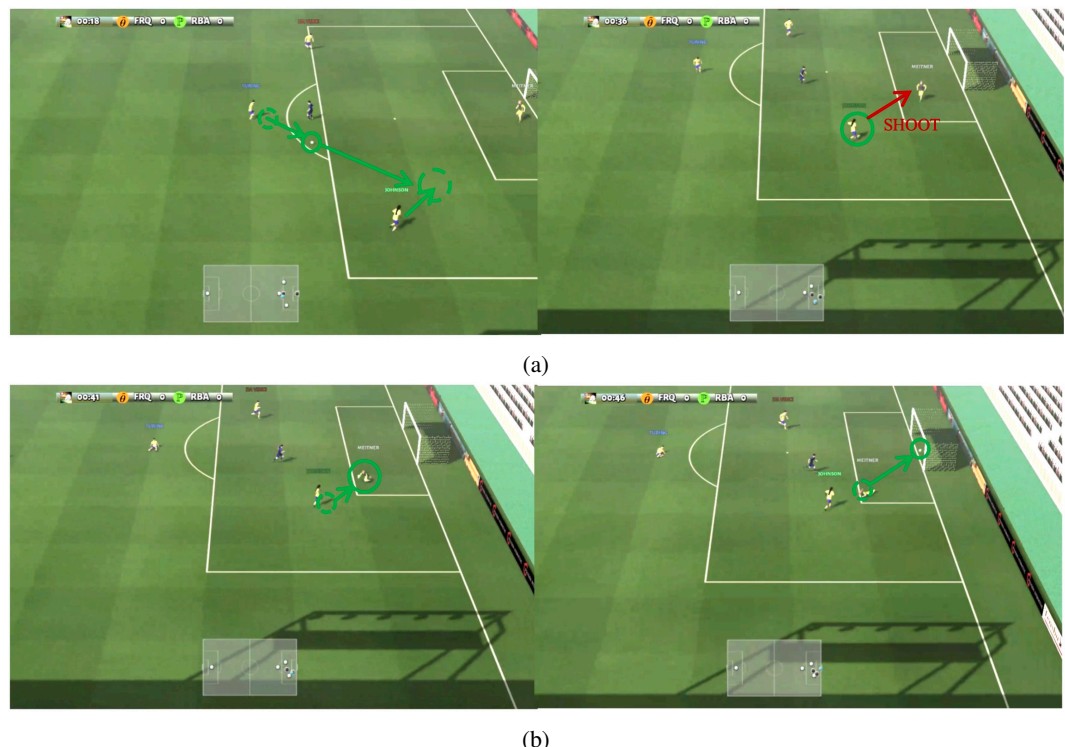

(a)

(b)

Figure 16: Visualization of the exploitation policy learned by EECE in the academy_3_vs_1_with_keeper scenario, showing agent behaviors at timesteps $t = 18$, 36, 41, and 46, including coordinated movements, passing, and shooting. Green arrows indicate the movements of the ball or teammates, while red arrows represent shooting actions. The figures highlight how EECE enables agents to perform high-level cooperative strategies, including effective passing, goal-oriented positioning and shooting.

the additional cost introduced by the ensemble models and exploration policy remains reasonable. Overall, EECE imposes only a marginal increase in computational requirements while maintaining competitive efficiency.

Table 7: Training modes across different baselines and environments.

| Environment | CDS | EMC | FOX | ICES | EECE |
|---|---|---|---|---|---|
| SMAC | episode | episode | episode | parallel | parallel |
| GRF | episode | episode | episode | episode | episode |

Table 8: Training batch_size across different baselines and environments.

| Environment | CDS | EMC | FOX | ICES | EECE |
|---|---|---|---|---|---|
| SMAC | 32 | 32 | 32 | 128 | 128 |
| GRF | 32 | 32 | 8 | 32 | 32 |

# I  TASK-RELEVANT COOPERATIVE INTRINSIC REWARD (EXTENSION)

While $r^{\mathrm{coo}}(s_t, \boldsymbol{a}_t)$ encourages the exploration of actions that jointly induce significant state changes, thereby effectively reducing the vastness of the exploration space, it does not guarantee that such

Table 9: Average training time (hours) across different baselines and environments.

| Environment | Scenario (T) | CDS | EMC | FOX | ICES | EECE |
|---|---|---|---|---|---|---|
| SMAC | 2s_vs_1sc (2M) | 9.0 | 17.6 | 11.1 | 3.1 | 3.4 |
| | 3m (2M) | 13.1 | 18.7 | 53.2 | 5.9 | 4.6 |
| | 8m (2M) | 29.4 | 20.0 | 64.8 | 5.5 | 5.6 |
| | 2s3z (2M) | 15.2 | 20.2 | 28.6 | 4.4 | 4.4 |
| | 1c3s5z (2M) | 25.5 | 22.3 | 38.5 | 6.6 | 4.9 |
| | 5m_vs_6m (3M) | 19.0 | 28.2 | 109.5 | 9.2 | 8.5 |
| GRF | academy_3_vs_1_with_keeper (4M) | 21.3 | 81.7 | 18.8 | 18.3 | 35.3 |
| | academy_corner (4M) | 24.4 | 76.3 | 29.5 | 22.5 | 32.6 |
| | academy_counterattack_hard (4M) | 27.0 | 78.7 | 24.4 | 22.3 | 33.2 |

cooperative behaviors are aligned with the task objective. Whether the discovered cooperative patterns are indeed task-relevant must ultimately be judged by the external reward provided by the environment.

As an optional extension, the cooperative reward can be augmented with the sum of each agent's counterfactual task reward, defined as

$$r^i(s_t, a_t^i \mid \boldsymbol{a}_t^{-i}) = r(s_t, \boldsymbol{a}_t) - r(s_t, \boldsymbol{a}_t^{-i}),$$

which measures agent $i$'s marginal contribution to the task reward under the current joint action. This counterfactual perspective allows the cooperative exploration to remain both coordinated and task-relevant. Formally, the enhanced cooperative reward is given by:

$$r^{\text{coo}}(s_t, \boldsymbol{a}_t) = \sum_{i=1}^n I\big(a_t^i; \delta s_{t+1} \mid s_t, \boldsymbol{a}_t^{-i}\big) + \lambda \sum_{i=1}^n r^i(s_t, a_t^i \mid \boldsymbol{a}_t^{-i}), \tag{25}$$

where the counterfactual baseline $r(s_t, \boldsymbol{a}_t^{-i})$ can be approximated using deep ensemble dynamics models.

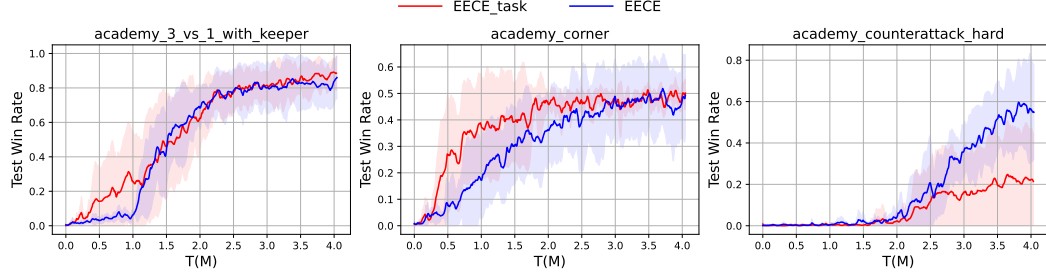

Figure 17: Performance of EECE with task-relevant cooperative reward in GRF scenarios.

Figure 17 reports the performance of EECE augmented with the task-relevant cooperative reward (denoted as EECE_task with $\lambda = 1$) on GRF tasks. We observe that in `academy_3_vs_1_with_keeper` and `academy_corner`, EECE_task converges faster compared to the baseline. However, in `academy_counterattack_hard`, its performance degrades. This may stem from the fact that under sparse reward settings, predicted rewards can be inaccurate, and incorporating such terms may introduce instability.

Overall, incorporating task-aware cooperative signals appears promising but remains challenging. Future extensions could leverage more principled approaches, such as goal-conditioned RL (Nasiriany et al., 2019; Na & Moon, 2024) or episodic control techniques (Pritzel et al., 2017; Lin et al., 2018), to construct more reliable task-relevant cooperative rewards. We leave this direction as a promising avenue for future research.

## J  LLM USAGE

In this work, large language models (LLMs) were employed primarily to assist in polishing the writing of this manuscript. The use of LLMs was limited to language refinement, ensuring clarity and readability, without influencing the technical content, experimental design, or conclusions.

## K  JUSTIFICATION OF PREDICTING STATE VARIANCE VS. PREDICTING NEXT STATE

*Proof.* **Expected Variance Consistency.** For clarity, we denote the ensemble of $K$ models predicting the next state or state variance as

$$s_{t+1}^{(k)} = f_k(s_t, \mathbf{a}_t) \ or \ s_{t+1}^{(k)} - s_t = f_k(s_t, \mathbf{a}_t) \quad k = 1, \ldots, K.$$

**Case 1: Predicting $s_{t+1}$ directly.** The ensemble mean and variance are

$$\mu(s_t, \mathbf{a}_t) = \frac{1}{K} \sum_{k=1}^{K} s_{t+1}^{(k)} = s_{t+1},$$

$$\sigma^2(s_t, \mathbf{a}_t) = \frac{1}{K} \sum_{k=1}^{K} \left\| s_{t+1}^{(k)} - \mu(s_t, \mathbf{a}_t) \right\|^2 = \frac{1}{K} \sum_{k=1}^{K} \left\| s_{t+1}^{(k)} - s_{t+1} \right\|^2.$$

**Case 2: Predicting the variance $\delta s_{t+1} = s_{t+1} - s_t$.** The ensemble mean and variance of the variance predictions are

$$\mu(s_t, \mathbf{a}_t) = \frac{1}{K} \sum_{k=1}^{K} \delta s_{t+1}^{(k)} = s_{t+1} - s_t,$$

$$\sigma^2(s_t, \mathbf{a}_t) = \frac{1}{K} \sum_{k=1}^{K} \left\| \delta s_{t+1}^{(k)} - \mu(s_t, \mathbf{a}_t) \right\|^2 = \frac{1}{K} \sum_{k=1}^{K} \left\| \delta s_{t+1}^{(k)} - (s_{t+1} - s_t) \right\|^2.$$

Since $s_t$ is known and fixed as input to the model, we have

$$\delta s_{t+1}^{(k)} - (s_{t+1} - s_t) = (s_{t+1}^{(k)} - s_t) - (s_{t+1} - s_t) = s_{t+1}^{(k)} - s_{t+1}.$$

Thus, the variance of the variance predictions equals the variance of the original next-state predictions:

$$\sigma^2(s_t, \mathbf{a}_t) = \frac{1}{K} \sum_{k=1}^{K} \left\| s_{t+1}^{(k)} - s_{t+1} \right\|^2.$$

So, predicting the variance $\delta s_{t+1}$ or predicting $s_{t+1}$ directly leads to the same expected variance. Hence, the expected ensemble variance is unchanged by this reparameterization. $\square$

## L  RELATED WORKS

Exploration remains a fundamental challenge in both single-agent RL and MARL. In MARL, the exponentially large joint state action space and the need for cooperative behaviors make effective exploration particularly difficult. Existing approaches can be broadly grouped into three families.

**Diversity-driven exploration.** A dominant line of work encourages agents to diversify their behaviors. MAVEN (Mahajan et al., 2019) introduces a latent variable to modulate the joint policy and induce diverse modes. CDS (Li et al., 2021) maximizes the mutual information between agent identities and their trajectories to enlarge the coverage of visited states. EMC (Zheng et al., 2021) uses prediction-error based novelty signals from individual Q-networks. MACDE (Xu & Kaneko, 2023) extends ICM, a curiosity-driven exploration method for single-agent environments, to the multi-agent setting. ADER (Kim & Sung, 2023) extends SAC by assigning agent-specific entropy coefficients. FOX (Jo et al., 2024) promotes diversity through formation-level novelty. These methods largely rely on reducing state dimensionality or constructing diversity metrics (e.g., trajectory-identity MI, formation counts) to encourage broader state-space visitation.

**Cooperative exploration.** Another stream aims to model how agents influence each other and the environment, thereby promoting coordinated exploration. EITI/EDTI (Wang et al., 2019) estimate how one agent's behavior affects another agent's transitions to guide exploration toward critical states. LAIES (Liu et al., 2023) tackles the "lazy-agent" phenomenon by building causal graphs that quantify agent diligence. ICES (Li et al., 2024c) decomposes latent environmental transitions to estimate each agent's independent contribution. These methods derive cooperation-oriented exploration signals from inter-agent or agent-environment interaction structures, often inspired by real-world collaborative processes.

**Goal-oriented exploration.** A third line leverages goal-conditioned ideas to generate meaningful exploration targets. MASER (Jeon et al., 2022) constructs sub-goals from experience. CMAE (Liu et al., 2021) builds shared goals within a constrained space. PMIC (Li et al., 2022) employs mutual information to design intrinsic rewards that help agents escape suboptimal collaboration patterns. LAGMA (Na & Moon, 2024) embeds goal values in a latent space and integrates GCRL into MARL. These approaches use replay experience to identify high-value regions and guide agents through goal-conditioned reward shaping.

**Limitations of prior exploration methods.** Existing exploration techniques share two major limitations. Regardless of the category, these methods require generating stable and reliable signals in high-dimensional state–action spaces. However, approaches based on a single predictive model often fail to provide trustworthy predictions of state transitions in complex dynamics: model errors and overfitting can directly lead to fluctuations in the exploration signal, thereby degrading exploration quality (Pathak et al., 2019; Zheng et al., 2021; Xu & Kaneko, 2023; Liu et al., 2021; Jeon et al., 2022). Second, prior work typically focuses on either diversity or cooperation in isolation, lacking a unifying perspective that jointly captures state novelty and multi-agent cooperative structure. This separation makes it difficult to exploit the complementary nature of these two forms of exploratory guidance, especially in complex cooperative tasks. These limitations motivate our ensemble-based unified exploration framework, which leverages multiple predictive models to provide reliable uncertainty estimates and jointly integrate diversity-driven and cooperation-driven exploration.

**Ensemble models in RL.** Deep ensembles have been widely adopted in RL as a stable tool for uncertainty estimation (Lakshminarayanan et al., 2017). Broadly, their usage can be divided into two main directions. The first direction is *value ensembles*, which build ensembles of Q-functions, replacing a single Q-value output with multiple predictions to capture uncertainty in value estimation. Representative works include SUNRISE (Lee et al., 2021b), MeanQ (Liang et al., 2022), EDE (Jiang et al., 2023), and CeSD (Bai et al., 2024). More recently, value ensembles have also been extended to MARL, e.g., EMAX (Schäfer et al., 2023). It is important to note that value ensembles only introduce the ensemble concept into value function estimation and do not explicitly leverage environment dynamics. As a result, the learning process primarily fits the observed data distribution and captures value bias, without extracting additional structural information from environment feedback. The second direction leverages ensembles of *environment dynamics models*, i.e., world models, to estimate uncertainty over state transitions. By modeling the environment dynamics, these ensembles can directly exploit feedback from the environment and capture epistemic uncertainty through disagreement among predictions. Such methods can generate more reliable exploration signals, for instance, using intrinsic rewards based on information gain or prediction disagreement (Pathak et al., 2019; Sekar et al., 2020; Sukhija et al., 2024).

## Our Contributions

*Challenge.* In this work, we address the challenge of efficient and reliable exploration in multi-agent reinforcement learning, where high-dimensional state-action spaces and complex agent interactions introduce significant complexity, often leading to unreliable reward signals. Relying solely on diversity-driven or cooperation-driven exploration methods is often insufficient.

*Method.* To tackle these challenges, we integrate ensemble-based environment models into the MARL framework to obtain stable and reliable uncertainty estimates, which serve as the basis for diversity-driven exploration by introducing the concept of information gain into MARL. We further introduce a novel mutual-information-based approach that leverages the ensemble as a proxy to compute cooperation-oriented rewards. By disentangling the influence of individual agent actions on global state transitions, our method quantifies each agent's cooperative contribution, improving both

the efficiency and stability of exploration. Building on these components, we propose a dynamic weighting mechanism and a dual-policy framework. The dynamic weighting mechanism adaptively balances diversity-driven and cooperation-driven rewards throughout training, while the dual-policy framework separates exploration and main policies, mitigating reward interference and enabling the two exploration signals to complement and reinforce each other.

*Uniqueness.* Unlike prior approaches that treat diversity- and cooperation-driven exploration independently, our framework delivers a unified and scalable solution that integrates ensemble models, intrinsic rewards for diversity based on information gain, mutual-information-based cooperation rewards, dynamic weighting, and dual-policy learning. This integration substantially enhances exploration efficiency, stability, and coordination in high-dimensional multi-agent environments. Crucially, our approach brings ensemble models into MARL and leverages information-theoretic measures to effectively and organically fuse diversity-driven and cooperation-driven exploration, rather than simply combining the two strategies.

*Future Direction.* In the future, our framework can be extended to incorporate goal-oriented exploration, enabling the generation of targeted cooperative signals that more accurately reflect realistic collaborative scenarios.

## M  EFFECT OF $K$

To evaluate the effect of different ensemble sizes $K$ on agent performance, we conducted experiments with $K = 3, 5, 8, 10$ in the `2s_vs_1sc` scenario. As shown in Figure 18, a smaller ensemble size ($K = 3$) results in slightly slower convergence, while $K = 5$ already provides stable and reliable learning performance. Increasing $K$ beyond 5 (i.e., $K = 8$ or $K = 10$) does not yield significant improvements, indicating that $K = 5$ offers a good trade-off between uncertainty estimation quality and computational cost.

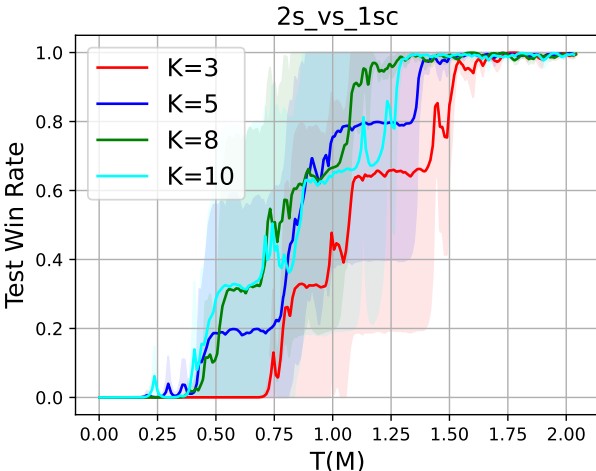

Figure 18: Performance comparison with different ensemble sizes $K$ on the `2s_vs_1sc` scenario.

