# OpenReview forum: "EECE: Ensemble-based Epistemic and Cooperative Exploration for Multi-Agent Reinforcement Learning"
_ICLR.cc/2026/Conference — ICLR 2026 Conference Withdrawn Submission_

### Official Review · Reviewer_e8pD · 2025-10-26

**Soundness:** 3
**Presentation:** 3
**Contribution:** 2
**Rating:** 4
**Confidence:** 4

**Summary:**

This work introduces a new framework called Ensemble-based Epistemic and Cooperative Exploration (EECE), designed to solve a key challenge in multi-agent reinforcement learning (MARL): efficient exploration in complex, cooperative tasks with sparse rewards. EECE uses an ensemble dynamics model to simultaneously encourage both. It creates two distinct information-theoretic intrinsic rewards, including an epistemic signal that rewards agents for exploring transitions with high uncertainty and a cooperative signal that rewards agents for working together to create changes in the global state, quantified using mutual information. EECE achieves some improvements in both exploration efficiency and final performance.

**Strengths:**

1. The authors propose EECE, a single, unified framework that addresses two critical and often separate challenges in MARL: encouraging effective exploration and promoting inter-agent cooperation.

2. EECE uses information gain to quantify uncertainty, guiding agents toward truly novel states (directional exploration) rather than just random exploration. EECE Uses mutual information to measure each agent's influence on the global state. This reward is aggregated to encourage collaborative, proactive behaviors. Both the epistemic and cooperative rewards are derived directly from the same ensemble of dynamics models.

3. This work develops a novel dual-policy mechanism where exploration and exploitation policies are learned independently. This helps stabilize the learning process by mitigating non-stationarity and credit assignment problems that intrinsic rewards can often cause.

**Weaknesses:**

1. The authors claim that previous methods encouraging either diversity or cooperation often fail to provide reliable exploration signals in high-dimensional state-action spaces and lack a unified treatment of both aspects, which limits their overall effectiveness. However, there is no theoretical or empirical evidence provided to substantiate the authors' claims. I suggest that the authors add more discussion on the limitations they claim to strengthen the contributions of this work.

2. EECE needs a reliable model of environment dynamics to support epistemic and cooperative exploration. However, the dynamics of high-dimensional multi-agent systems, especially with a large number of agents, are highly complicated. The convergence of the learning of the dynamic model should be addressed. Moreover, an ensemble prediction may require training $K$ deep neural networks. The computational complexity can be high. The discussion of the effects of different values of $K$ on the performance of agents is also missing.

3. The definition of cooperation as "the coordinated effort of all agents to collectively induce variations in the global state, with the actions of individual agents contributing to these changes," is confusing. These variations cannot be referred to as cooperation because non-cooperative behaviors among agents can also lead to variations. I suggest the authors reconsider the concept of cooperative exploration proposed in this work.

4. The dual-policy strategy seems to significantly improve the performance. However, if EECE uses the previous combining methods, the ablation results show limited performance improvement, which demonstrates the high dependency between dual-policy learning and the two proposed exploration methods. Why does the necessary dual-policy strategy improve the proposed exploration methods?

**Questions:**

Please see the weaknesses above.

---

> ### Author Response · Authors · 2025-11-19
> **Response to Reviewer e8pD(Part 1)**
>
> We sincerely thank the reviewer for their thoughtful feedback and valuable time spent reviewing our work.
>
> **W1-more discussion on the limitations** We sincerely thank the reviewer for this insightful comment. While key relevant studies and the problems addressed by our work were discussed in the **second and third paragraphs of the Introduction** and **Section 4 RELATED WORKS**, we recognized that a more detailed discussion on the limitations of existing methods was needed. In response, we have reorganized and expanded the related work section, categorizing prior methods and explicitly highlighting the gaps our study addresses: unstable exploration signals in high-dimensional state-action spaces and the lack of a unified framework integrating diversity-driven and cooperation-driven exploration. We believe these additions, now available in **Appendix L RELATED WORKS**, make our contributions and motivations significantly clearer and strengthen the overall presentation and contribution of the paper.
>
> **W2—The convergence of the dynamic model and computational complexity.** We thank the reviewer for raising this point. In RL and MARL, it is common to introduce sample-based predictive modules to aid exploration, such as **RND [1], ICM [2], Disagreement [3], and Ensembles [4]**. Learning a dynamic model is a widely adopted approach in both model-based and model-free RL. Specifically, ensemble methods provide an effective mechanism to quantify epistemic uncertainty. As a component supporting exploration, the dynamic model gradually converges during training, similar to predictors used in **RNG** [1] or **ICM** [2]. Unlike these methods, ensembles additionally provide uncertainty estimates, making them more robust and better suited to stochastic environments. The choice of $K$ impacts computational efficiency. In our experiments, we adopt ,$K$ consistent with prior literature [3-4], which provides sufficiently accurate uncertainty estimates while balancing computational cost . While the computational cost does increase with the number of agents, our results show that the training time remains manageable. A detailed discussion of training time for all considered algorithms is provided in **Appendix H TRAINING TIME**.
>
> To address the reviewer’s concern and enrich the paper, we conducted experiments with different ensemble sizes $K$ ($K=3,5,8,10$). The results show that $K=3$ converges slightly slower, while $K=5$ already provides stable and reliable performance. Increasing $K$ further to 8 or 10 does not bring significant improvement. These findings confirm that our choice of $K=5$ strikes a good balance between computational cost and accurate uncertainty estimation. Detailed results and analysis can be found in **Appendix M EFFECT OF K**, which will be moved to the appropriate section in the final version.
>
> **W3 – Concept of Cooperative Exploration** We sincerely thank the reviewer for the valuable comment regarding the concept of “cooperative exploration.” In multi-agent systems, the notion of cooperation is not straightforward to define. Current methods often rely on intuitive heuristics to encourage cooperative behaviors. For example, EITI/EDTI [6] models how one agent’s state-action affects another agent’s state transitions, considering large mutual influence as a form of cooperation; LAIES [7] addresses the “lazy agent” problem using causal graphs to quantify agent diligence, where higher diligence is interpreted as cooperative behavior; ICES [5] measures an agent’s independent contribution to state-transition latent variables, interpreting higher contributions as cooperation. Similarly, in our work, we define cooperative exploration as *“the coordinated effort of all agents to collectively induce variations in the global state, with the actions of individual agents contributing to these changes.”*
>
> While these computation rules are not strict necessary and sufficient conditions for cooperation, they capture intuitively necessary aspects, and the corresponding intrinsic rewards effectively reduce the exploration space and improve exploration efficiency. We have also considered that non-cooperative behaviors among agents may also lead to state variations. As discussed in **Limitations and Future Works**, *“in realistic settings, effective cooperation is often task-driven, and exploring task-aware cooperative rewards is a promising direction, for example via goal-conditioned RL or episodic control techniques”*, and we conducted preliminary experiments in **Appendix I TASK-RELEVANT COOPERATIVE INTRINSIC REWARD (EXTENSION)**. However, these directions go beyond the core contributions of this paper and are therefore left for future work. More refined definitions and formalizations of cooperative behavior remain an open problem for the community.

---

> ### Author Response · Authors · 2025-11-19
> **Response to Reviewer e8pD(Part 2)**
>
> **W4-Dual-Policy Strategy.** In standard RL, a common practice is to incorporate intrinsic rewards directly into the external rewards, e.g.,$r^{\text{tot}} = r^{\text{ext}} + 0.5 r^{\text{int}}$. However, this approach requires careful tuning of the intrinsic reward coefficient to balance exploration and exploitation, which is often tedious and unstable [4, 5]. In MARL, this issue is further amplified. Agents' policies keep changing during training, and the environment typically provides only a global reward. These factors exacerbate non-stationarity and credit-assignment difficulties. Value-decomposition methods such as QMIX are specifically designed to infer individual utility from global rewards. Directly adding intrinsic rewards to extrinsic rewards introduces an additional reward component that must also be decomposed, making coefficient tuning even harder and increasing non-stationarity and credit-assignment issues, ultimately degrading performance. To address these challenges, we develop a dual-policy mechanism where exploration and exploitation policies are trained independently. Both policies use value-decomposition architectures, which avoids the issues caused by mixing intrinsic and extrinsic rewards and allows the model to learn the long-term value of exploratory actions. This design makes training more stable and leads to clear performance improvements.
>
>
>
> [1] Exploration by Random Network Distillation. ICLR 2019
>
> [2] Curiosity-driven Exploration by Self-supervised Prediction. ICML 2017
>
> [3] Self-supervised exploration via disagreement.ICML 2019
>
> [4] Maxinforl: Boosting exploration in reinforcement learning through information gain maximization. ICLR 2025.
>
> [5] Individual Contributions as Intrinsic Exploration Scaffolds for Multi-agent Reinforcement Learning, ICML 2024
>
> [6] Influence-based multi-agent exploration , ICLR 2020
>
> [7] Lazy agents: a new perspective on solving sparse reward problem in multi-agent reinforcement learning, ICML 2023

---

### Official Review · Reviewer_yUUJ · 2025-10-28

**Soundness:** 3
**Presentation:** 3
**Contribution:** 4
**Rating:** 6
**Confidence:** 4

**Summary:**

The paper proposes Ensemble-Based Epistemic and Cooperative Exploration (EECE), a framework that promotes
both exploration and cooperation in sparse-reward, partially observable environments. It combines information
gain and conditional mutual information based intrinsic rewards through a dynamic weighting scheme, and trains
two policies (exploration and exploitation) per agent. The former learns from intrinsic rewards, the latter extrinsic
rewards. Agents actions are sampled from the two policies through a dual-policy mechanism.

The paper makes two main contributions. Firstly, the authors propose a solution to address the lack of a method
that accomplishes both diverse exploration and cooperation amongst agents. Secondly, they point out the
detriments of naively combining intrinsic and extrinsic rewards and introduce a novel dual-policy scheme that
effectively utilises these rewards separately. They also highlight the effectiveness of using information theoretics,
through their construction of the to promote coordination.

**Strengths:**

The application of the ensemble-based exploration method to the MARL domain and introduction of the
cooperation intrinsic reward constructed using conditional mutual information is a nice idea.

In contrast to existing methods that combine intrinsic and extrinsic rewards using weights, the authors used these rewards separately to train two policies per agent. The motivations for this framework, basis for each EECE component, and its implementation are
well communicated through clear writing and effective diagrams. Enough experimental details are provided for
reproducibility and the workings of the algorithm are clearly documented.

The claim that EECE promotes both diverse exploration and cooperation is mostly well supported. This is
demonstrated through the superior performance of EECE compared to state-of-the-art baselines. The
contributions of each key component of EECE is also quantified through rigorous ablation studies, demonstrating
soundness in the adopted principles for component construction. Considering these aspects along with clarity
and significance, this paper is of good quality.

**Weaknesses:**

The authors justify their choice of using the ensemble to predict state variance instead of $s_{t+1}$ by stating that
epistemic uncertainty is preserved as the expected variance is unaffected. Either than this note, no further
justifications or proofs were provided. Furthermore, although the provided figures visualising exploration policy
does demonstrate emergent cooperative behavior, no other results were included. Given the claims that EECE
promotes cooperation, perhaps this is insufficient.

The authors outlined that the dynamic weighting strategy adopted shifts focus from epistemic exploration to coordination over time but don't provide justification as to why this is effective.

* It is not clear why the authors chose to compare EECE with QMIX in section 5.3 to demonstrate that EECE promotes diverse exploration.

* Some highly relevant works aren't discussed or compared against e.g. [1].

* It would be beneficial to include motivations for the design and purpose of the adopted dynamic weighting
strategy.

* Although the effects of varying the parameter has been investigated, no information was provided
regarding how this hyper parameter should be tuned. Provision of some guidance regarding this or its
inclusion for future work would improve the paper.

* Relying solely on visualisations of agents’ exploration policies might be insufficient to prove that EECE
promotes coordination. Inclusion of qualitative metrics to complement these visualisations, such as the
measured mutual information between agents’ actions, would be more convincing.

[1] Ensemble Value Functions for Efficient Exploration in Multi-Agent Reinforcement Learning. Schäfer et al. AAMAS 2025.

**Questions:**

1. Can you justify why predicting state variance has no effect on the expected variance observed from predicting the next state itself?

2. Can you explain why you chose to compare EECE with QMIX to demonstrate that EECE promotes
exploratory behaviours?

---

> ### Author Response · Authors · 2025-11-19
> **Response to Reviewer yUUJ (Part 1)**
>
> We sincerely thank the reviewer for this insightful and constructive comment, which has helped clarify and strengthen our work.
>
> **Q1-predicting state variance**  As this point is not central to our paper, we did not include a detailed proof in the main text. Here, we provide a clarification. Predicting the next state is equivalent to predicting the residual $\delta s_{t+1} = s^{k}_{t+1} - s_t = f_k(s_t, \boldsymbol{a}_t)$, similar to a residual network.
>
> Assume that the predictive model has already been trained.
>
> When the model predicts $s_{t+1}$ directly:
>
> $\mu(s\_t, \boldsymbol{a}\_t) = \frac{1}{K}\sum_{k=1}^{K} s\_{t+1}^{(k)} = s\_{t+1}$,$\sigma^2(s\_t, \boldsymbol{a}\_t) = \frac{1}{K}\sum\_{k=1}^{K}\big\|s\_{t+1}^{(k)} - \mu(s\_t, \boldsymbol{a}\_t)\big\|^2 = \frac{1}{K}\sum\_{k=1}^{K}\big\|s\_{t+1}^{(k)} - s\_{t+1}\big\|^2$
>
> When the model predicts the residual $\delta s_{t+1} = s_{t+1} - s_t$:
>
> $\mu(s\_t, \boldsymbol{a}\_t) = \frac{1}{K}\sum\_{k=1}^{K} \delta s\_{t+1}^{(k)} = s\_{t+1} -s\_{t}$,   $\sigma^2(s\_t, \boldsymbol{a}\_t) = \frac{1}{K}\sum\_{k=1}^{K}\big\|\delta s\_{t+1}^{(k)} - \mu(s\_t, \boldsymbol{a}\_t)\big\|^2 = \frac{1}{K}\sum\_{k=1}^{K}\big\|\delta s\_{t+1}^{(k)} - [s\_{t+1}-s\_{t}]\big\|^2  $
>
> Since $s_t$ is known and used as the model input, we have:$\sigma^2(s\_t, \boldsymbol{a}\_t) =\frac{1}{K}\sum\_{k=1}^{K}\big\|s\_{t+1}^{(k)}-s\_t - [s\_{t+1}-s\_{t}]\big\|^2 =  \frac{1}{K}\sum\_{k=1}^{K}\big\|s\_{t+1}^{(k)} - s\_{t+1}\big\|^2 $
>
> Therefore, the expected variance is identical whether we predict $s_{t+1}$ directly or the residual $\delta s_{t+1}$. We have temporarily added the detailed justification to **Appendix K: JUSTIFICATION OF PREDICTING STATE VARIANCE VS. PREDICTING NEXT STATE**, and it will be moved to an appropriate location in the next version.
>
> **Q2-chose to compare EECE with QMIX to demonstrate that EECE promotes exploratory behaviours** We chose QMIX as the baseline in Section 5.3 because it is a widely adopted and representative value-decomposition method in MARL that does not incorporate explicit exploration mechanisms beyond standard ε-greedy. By comparing EECE with QMIX, we can clearly isolate and demonstrate the benefits of our proposed exploration framework. Specifically, EECE encourages exploration through epistemic uncertainty estimation and cooperation-driven intrinsic rewards, which are absent in QMIX. This comparison thus directly highlights how EECE enhances exploratory behaviors, validating the effectiveness of our approach.
>
> **W1 – Validation of cooperative behavior. Relying solely on visualisations of agents’ exploration policies might be insufficient.** For diversity-driven exploration, the effect of EECE can be quantitatively evaluated. For example, in **Figure 7** and **Figure 14**, it is clear that EECE visits a larger portion of the state space compared to QMIX. However, for cooperative behavior, there is currently no straightforward quantitative metric. In this work, we adopt the measure defined by **“the sum of these marginal influences provides a measure of the overall level of cooperation for a joint action”** (Equation 7). This encourages the exploration policy to select joint actions that actively change the environment, which reflects a general characteristic of cooperative actions. Essentially, this reward compresses the effective exploration space to improve exploration efficiency. Even if one were to measure the mutual information between agents’ actions, it may not fully capture cooperation. This is because, as described in **Appendix I: TASK-RELEVANT COOPERATIVE INTRINSIC REWARD (EXTENSION)**, real-world cooperation is inherently goal-directed, and strong mutual influence between agents may merely represent interference rather than genuine collaboration. Further research is needed to develop better quantitative metrics. Currently, most studies use visualization to demonstrate the effect of cooperative exploration. In our work, Figures **15** and **16** in Section 5.3 illustrate that EECE has a positive influence on cooperative behavior. Notably, in **Figure 15**, the exploration policy recommends moving closer to the enemy in the first three time steps, showing that it generates actions with long-term exploratory value. Moreover, two agents performing complementary actions, one moving and one attacking, demonstrates a form of cooperation that is crucial for success in this game. At present, fully quantifying such cooperative behavior is not feasible. Even if we show that the exploration policy obtains cooperative rewards (i.e., mutual information between actions and state transitions), this only indicates that the RL algorithm can learn high-reward actions. We hope the reviewer understands this limitation.

---

> ### Author Response · Authors · 2025-11-19
> **Response to Reviewer yUUJ (Part 2)**
>
> **W2 – Role of the dynamic weighting strategy.** The dynamic weighting strategy adopted in our work is designed to gradually shift the exploration focus from purely epistemic uncertainty-driven exploration to multi-agent coordination as training progresses. The underlying intuition is that, during the early stages of training, uncertainty-driven exploration helps agents quickly discover potentially valuable regions of the state space, whereas later in training, improving cooperative capabilities becomes more critical for overall task performance in multi-agent settings. While this effect is difficult to formally prove, ablation studies demonstrate the positive impact of this strategy. Moreover, from **Figure 14** (Number of visited states and win rate of EECE and QMIX on *academy_3_vs_1_with_keeper*) and **Figure 7** (Number of visited states and win rate of EECE and QMIX on *2s_vs_1sc*), we observe that the number of visited state regions continues to increase, while the win rate shows a rapid improvement after a certain period of training. This suggests that the cooperation-driven reward effectively helps agents discover key actions, supporting the intended role of the dynamic weighting strategy.
>
> **W3 – Highly relevant works.** We thank the reviewer for this comment. Methods such as [1] and other Ensemble Value Function approaches train multiple Q-networks and use UCB-based strategies to select actions with high uncertainty. These methods focus solely on estimating uncertainty in Q-values and do not leverage environment transition information. In contrast, our work introduces ensemble methods in the form of environment dynamics models, which fully exploit transition information to provide exploration signals. As noted in our response to **Reviewer 6ZfR** and in references [14–18], we will further expand the discussion of related works to highlight these distinctions. We have expanded the related work section to include additional relevant studies, which can be found in **Appendix L RELATED WORKS**.
>
> [1] Ensemble Value Functions for Efficient Exploration in Multi-Agent Reinforcement Learning. Schäfer et al. AAMAS 2025.
>
> **W4 – Motivations for the dynamic weighting strategy.** We thank the reviewer for this comment. Due to page limitations in the initial submission, the **Dynamic Weighting Strategy** subsection only briefly introduced the scheme as: *“The scheme shifts from early epistemic exploration to later cooperative exploration, effectively balancing the two signals for efficient multi-agent learning.”* To provide further clarification, we have expanded this subsection with the following explanation: *“The dynamic weighting strategy is motivated by the evolving roles of exploration and cooperation during training. Early in training, agents have limited knowledge of the environment, and emphasizing epistemic rewards encourages visiting novel states and reducing model uncertainty. Later, as agents acquire sufficient information, cooperative behaviors become critical for collective performance. Gradually shifting the weight from epistemic to cooperative rewards allows EECE to first explore effectively and then focus on coordination, resulting in more efficient and stable multi-agent learning.”*
>
> **W5 – Hyperparameters.** We thank the reviewer for this suggestion. In **Appendix E: Additional Experimental Results**, we briefly analyze the influence and robustness of the hyperparameters. Our experiments show that, in GRF, smaller values of $\alpha_{\text{min}}$ and larger values of $\kappa$ consistently lead to improved performance, indicating that cooperation-driven exploration is more effective in these tasks. In contrast, for SMAC tasks, larger $\alpha_{\text{min}}$ and smaller $\kappa$ yield better results, suggesting that epistemic exploration plays a more dominant role. This demonstrates that hyperparameter tuning depends on the state space and cooperative characteristics of each environment. We also note in the **Appendix E** that *tuning these hyperparameters to the characteristics of each benchmark can further enhance overall performance*, and we discuss in **Limitations and Future Work** that *developing adaptive mechanisms to combine these two types of exploration rewards is an interesting avenue for future research*.We believe that these explanations and updates further improve the paper.

---

### Official Review · Reviewer_6ZfR · 2025-10-31

**Soundness:** 4
**Presentation:** 3
**Contribution:** 1
**Rating:** 4
**Confidence:** 5

**Summary:**

This paper introduces EECE, a framework for exploration in MARL under sparse rewards. The method leverages an ensemble of dynamics models to generate two information-theoretic intrinsic rewards: an epistemic reward for novelty-seeking and a cooperative reward for promoting coordinated actions. These rewards are integrated via a dynamic weighting strategy and a dual-policy mechanism, leading to performance gains over existing methods on challenging SMAC and GRF benchmarks.

**Strengths:**

1. The proposed method is well-designed and logically constructed. Each component is thoughtfully designed to contribute to the goal of encouraging both diverse and cooperative exploration.
2. The empirical evaluation is quite good. The authors have conducted thorough experiments on challenging and appropriate benchmarks (SMAC and GRF), and the comprehensive ablation studies effectively demonstrate the contribution of each part of the EECE framework. The qualitative analyses and visualizations are also very helpful in providing insight into the learned behaviors.

**Weaknesses:**

1. The exploration in MARL is a research topic that has been investigated by many previous research work. The novelty of this paper seems marginal. Specifically,
    - The paper would benefit from a more precise definition of the research gap it addresses. The stated goal of "enhancing diverse exploration and inter-agent cooperation" is a central theme in many prior works, including those cited. To better distinguish this work, it would be helpful to clarify paper's unique contribution.
    - Regarding the technical contributions, the authors have integrated several powerful techniques. However, these individual components—ensemble-based uncertainty estimation [1-4], information-theoretic rewards for cooperation [5], and dual-policy training architectures [6-7]—are established concepts in the literature. Consequently, the paper's primary contribution appears to be the specific synthesis of these ideas.
2. The literature review could be made more comprehensive by including and discussing several closely related works, such as [8-9].

Improvement Suggestions (less important)
1. The authors acknowledged the increased computational time in the main text and reported the increased training time in appendix H. From my opinion, the increased computational time to trade off the enhanced performance is acceptable. For the sake of completeness, it would be better to also include the test-time latency.


[1] "Constrained Ensemble Exploration for Unsupervised Skill Discovery", ICML 2024 \
[2] "SUNRISE: A Simple Unified Framework for Ensemble Learning in Deep Reinforcement Learning", ICML 2021 \
[3] "On the Importance of Exploration for Generalization in Reinforcement Learning", NeurIPS 2023 \
[4] "Ensemble Value Functions for Efficient Exploration in Multi-Agent Reinforcement Learning", AAMAS 2025 \
[5] "Influence-based multi-agent exploration", ICLR 2020 \
[6] "Individual Contributions as Intrinsic Exploration Scaffolds for Multi-agent Reinforcement Learning", ICML 2024 \
[7] "Cooperative exploration for multi-agent deep reinforcement learning", ICML 2021 \
[8] "Lazy agents: a new perspective on solving sparse reward problem in multi-agent reinforcement learning", ICML 2023  \
[9] "An adaptive entropy-regularization framework for multi-agent reinforcement learning", ICML 2019

**Questions:**

1. Could the authors please clarify the specific research gap this paper aims to address, beyond the general goal of combining diverse and cooperative exploration? What specific shortcomings in prior methods does EECE resolve?
2. In light of the prior work mentioned ([1-7]), could the authors elaborate on the primary technical novelty of EECE? Why is this a meaningful step forward compared to simply combining existing techniques?
3. (less important) Could the authors expand the related work section to more explicitly differentiate EECE from methods like [5-7] and discuss its relationship to other relevant works like [8-9]?
4. (less important) The appendix provides training time, which is helpful. For completeness, could the authors also report the test-time latency/computational overhead compared to baselines? This is particularly relevant given the use of an ensemble model.
5. (less important) The results on the selected SMAC maps are quite good, with EECE achieving near-100% win rates in many cases. Have the authors considered evaluating EECE on more challenging, modern benchmarks like SMACv2 to further demonstrate its robustness and scalability?

---

> ### Author Response · Authors · 2025-11-19
> **Response to Reviewer 6ZfR (Part 1)**
>
> We sincerely thank the reviewer for their time, effort, and valuable comments on our work.
>
> **W1-novelty** Exploration has long been a central topic in RL and MARL. Our work, EECE, focuses on efficient exploration in MARL, and the novelty of this paper is neither marginal nor incremental.
>
> In the MARL literature, numerous studies have investigated the exploration problem, including MAVEN [1], EITI/EDTI [2], MASER [3], CMAE [4], CDS [5], EMC [6], MACDE [7],PMIC [8], ADER [9], LAIES [10], FOX [11], ICES [12], and LAGMA [13]. Broadly speaking, these methods can be categorized into the following classes:
>
> - **Diversity-driven exploration (encouraging agents to visit more of the state space):** MAVEN [1] modulates the joint policy using latent variables. CDS [5] maximizes the mutual information between agent trajectories and their identities to promote diverse trajectories. EMC [6] designs novelty-based intrinsic rewards based on the prediction error of independent Q-values. MACDE [7] extends ICM, a curiosity-driven exploration method for single-agent environments, to the multi-agent setting. ADER [9] extends SAC’s adaptive entropy coefficients to MARL, assigning agent-specific entropy coefficients to encourage diverse exploration. FOX [11] encourages agents to explore diverse formations.
>
> - **Cooperation-driven exploration (promoting agent interactions and influence on the environment):** EITI/EDTI [2] models the influence of one agent’s state-action pair on another agent’s state transitions to guide agents into critical regions. LAIES [10] addresses the lazy-agent problem in MARL by introducing a causal graph to define agent diligence, thereby promoting cooperative behavior. ICES [12] estimates each agent’s independent contribution to latent variables of state transitions, encouraging agents to influence the environment.
>
> - **Experience- or goal-conditioned reward design (leveraging high-value trajectories):** MASER [3] generates subgoals from experience to guide agents toward target regions. CMAE [4] constructs shared goals within a restricted space. PMIC [8] designs intrinsic rewards using mutual information while considering both successful and unsuccessful trajectories, enabling agents to progressively break suboptimal collaborations and learn better ones. LAGMA [13] records goal values in latent space and incorporates goal-conditioned RL into MARL.
>
> From a problem-solving perspective, exploration in multi-agent reinforcement learning faces two major challenges. **First**, the high-dimensional state action space: resulting from many agents and the redundancy of numerous states and actions and often leads to unstable reward signals (EMC [6]). One natural approach to address this issue is to reduce the effective exploration space. Methods such as EMC [6], FOX [11], CMAE [4], and LAGMA [13] implement various strategies to achieve this. Broadly speaking, this “dimensionality reduction” encourages agents to focus on regions that are most informative for their understanding of the environment, while providing stable and reliable exploration signals. In our work, this is realized via ensemble-based information gain, which guides agents toward high information regions and produces consistent intrinsic rewards. **Second**, current exploration methods are often fragmented, with diversity-driven and cooperation-driven strategies treated separately. We argue that in multi-agent settings, these components should act jointly to achieve effective exploration. For example, directed exploration reduces the effective exploration space, while cooperation-oriented rewards further constrain the action space. This is particularly important in multi-agent systems: for instance, five agents each with ten actions result in $10^5$ joint actions, many of which are redundant and may even hinder learning correct policies (ADER [9]). Therefore, effectively combining cooperation-driven and diversity-driven exploration is critical for efficient MARL. **Finally**, as noted in **Limitations and Future Work**, integrating goal-conditioned exploration represents a promising direction, since true cooperation should involve goal-directed modifications of the environment. In **Appendix I: TASK-RELEVANT COOPERATIVE INTRINSIC REWARD (EXTENSION)**, we present preliminary experiments exploring counterfactual task rewards. We believe that effective multi-agent exploration must consider these factors collectively in order to achieve robust performance and long-term development.

---

> ### Author Response · Authors · 2025-11-19
> **Response to Reviewer 6ZfR (Part 2)**
>
> From a technical perspective, our contributions can be summarized as follows: **First**, Although ensemble-based uncertainty estimation [19] is an existing technique and has become a fundamental component for uncertainty assessment, we are, to the best of our knowledge, the first to introduce it into MARL in the form of an **environment transition model**. Unlike Ensemble Value Functions [14,15,16,17,18], which train multiple Q-networks and use UCB strategies to select high-uncertainty actions, these methods do not leverage environment transition information and focus solely on Q-value prediction uncertainty. In contrast, our approach introduces ensemble methods as **dynamics models**, inspired by [20,21,22], allowing stable prediction and uncertainty estimation in high-dimensional multi-agent state–action spaces. Moreover, information gain provides a solid theoretical foundation that supports directed diversity-driven exploration. **Second**, Beyond implementing ensemble-based epistemic exploration in MARL, we combine it with **cooperation-driven exploration**. Specifically, we extend ensemble-based diversity exploration to the multi-agent domain by using the ensemble model to compute **mutual-information-based cooperation rewards**. This differs from the information-theoretic rewards in EITI/EDTI [2], representing a simple yet elegant way to integrate cooperation into ensemble-based exploration. **Third**, Based on the characteristics of diversity-driven and cooperation-driven exploration, we design a **dynamic weighting strategy** that balances the contributions of epistemic and cooperative rewards throughout training. **Finally**, We adopt a **dual-policy mechanism** to separate exploitation and exploration policies, enabling effective utilization of the combined reward signal. Our dual-policy mechanism is conceptually similar to CMAE [4] and ICES [12], but differs in two aspects: compared with CMAE [4], we train the exploration policy solely with the combined exploration reward; compared with ICES [12], we employ two separate value decomposition methods. These modifications allow the exploration policy to select actions with long-term exploratory value. For instance, as shown in Figure 8, the exploration policy chooses actions that move closer to the enemy in the first three steps, even though these actions do not produce immediate rewards.
>
> In summary, we observe that MARL faces two fundamental challenges: its inherently large state–action space and the separation between diversity-driven and cooperation-driven exploration. To address these issues, we introduce an ensemble-based approach that generates reliable exploration signals in high-dimensional multi-agent environments, and we effectively integrate epistemic (diversity) exploration with cooperation-oriented exploration into a unified framework. Although some underlying techniques have been explored in prior work, our method does **not** simply combine existing components. Instead, we identify core limitations in MARL exploration and propose **theoretically grounded and novel solutions** that directly address these challenges. We believe this constitutes a meaningful contribution to the design of exploration rewards and exploration strategies in MARL, and provides new insights for the community.
>
> **W2-more comprehensive literature review**  We sincerely thank the reviewer for this helpful suggestion. In the original submission, our Introduction and Related Works sections covered several highly relevant studies, but they may not have fully conveyed the broader context of the problem we address and the contribution we make. We have now expanded the literature review to incorporate the key works discussed in this response, and the updated content is included in **Appendix L RELATED WORKS** to help readers more easily understand the positioning and core novelty of our method. We will later move this expanded literature review to a more appropriate location in the main text and may further refine the Introduction accordingly.
>
> **Test-time latency.** The inference-time latency of all compared methods, including QMIX, CDS, EMC, FOX, ICES, and our EECE, is essentially identical. All these algorithms are built upon the value-decomposition framework, and once training is completed, the execution-time policy evaluation follows the **same computation pipeline and incurs the same computational cost**. In typical RL/MARL settings, policy networks are shallow and lightweight (only a few layers), making test-time inference negligible compared to models in domains such as large language models.

---

> ### Author Response · Authors · 2025-11-19
> **Response to Reviewer 6ZfR (Part 3)**
>
> **Q1-research gap， Q2-technical novelty**  Please refer to **W1–novelty** for a detailed explanation. We have also expanded **Appendix L RELATED WORKS** to include additional literature, a clearer discussion of the **limitations of prior exploration methods**, and a concise summary of **our contributions**. We kindly invite the reviewer to refer to this section for a comprehensive clarification.
>
> **Q3-expand the related work**  We thank the reviewer for this valuable suggestion. We have expanded the related work section to include additional relevant studies, which can be found in **Appendix L RELATED WORKS**.
>
> **Q4-test-time latency** Please refer to the **Test-time latency** for a detailed discussion.
>
> **Q5-SMACv2** The current work does not include evaluations on SMACv2, and incorporating this environment would require a substantial amount of additional time and effort. We hope the reviewer can understand this limitation. All baseline comparisons are conducted on SMAC and GRF, which we believe already provide a comprehensive evaluation. Looking ahead, we are eager to extend our framework to SMACv2 in future studies, particularly when exploring the integration of our approach with goal-conditioned RL.
>
>
>
> [1] Maven: Multi-agent  variational exploration, NIPS 2019
>
> [2] Influence-based multi-agent exploration , ICLR 2020
>
> [3] Maser: Multi-agent reinforcement learning with subgoals generated from experience replay buffer,ICML 2020
>
> [4] Cooperative exploration for multi-agent deep reinforcement learning, ICML 2021
>
> [5] Celebrating diversity in shared multi-agent reinforcement learning, NIPS 2021
>
> [6] Episodic multi-agent reinforcement learning with curiosity-driven exploration, NIPS 2021
>
> [7] Curiosity-driven Exploration for Cooperative Multi-Agent Reinforcement Learning, IJCNN 2023
>
> [8] Pmic: Improving multi-agent reinforcement learning with progressive mutual information collaboration, ICML 2022
>
> [9] An adaptive entropy-regularization framework for multi-agent reinforcement learning, **ICML 2023**
>
> [10] Lazy agents: a new perspective on solving sparse reward problem in multi-agent reinforcement learning, ICML 2023
>
> [11] Fox: Formation-aware exploration in multi-agent reinforcement learning, AAAI 2024
>
> [12] Individual Contributions as Intrinsic Exploration Scaffolds for Multi-agent Reinforcement Learning, ICML 2024
>
> [13] LAGMA: Latent goal-guided multi-agent reinforcement learning, ICML 2024
>
>
>
> [14] Constrained Ensemble Exploration for Unsupervised Skill Discovery, ICML 2024
>
> [15] On the Importance of Exploration for Generalization in Reinforcement Learning, NeurIPS 2023
>
> [16] SUNRISE: A Simple Unified Framework for Ensemble Learning in Deep Reinforcement Learning, ICML 2021
>
> [17] Reducing variance in temporal-difference value estimation via ensemble of deep networks, ICML 2022
>
> [18] Ensemble Value Functions for Efficient Exploration in Multi-Agent Reinforcement Learning, AAMAS 2025
>
>
>
> [19] Simple and scalable predictive uncertainty estimation using deep ensembles. NIPS 2017
>
> [20] Self-supervised exploration via disagreement.ICML 2019
>
> [21] Planning to explore via self-supervised world models, ICML 2020
>
> [22] Maxinforl: Boosting exploration in reinforcement learning through information gain maximization. ICLR 2025.

---

### Author Response · Authors · 2025-12-01
**Summary comment to AC (part 2/2)**

### Roles and Justifications of Method Components

1. **Predicting State Variance**

   Reviewer yUUJ suggested adding formal justification for predicting state variance rather than the next state. We have provided detailed explanations and included **Appendix K: JUSTIFICATION OF PREDICTING STATE VARIANCE VS. PREDICTING NEXT STATE** to clarify the rationale and theoretical grounding of this design choice.

2. **Concept of Cooperative Exploration**

   Reviewer e8pD noted that non-cooperative behaviors can also affect the global state. It should be emphasized that in multi-agent systems, the notion of cooperation is not straightforward to define. Existing methods typically use intuitive heuristics to encourage cooperative behavior. In our work, the cooperative exploration reward can be understood as reducing the joint action exploration space to those combinations that jointly induce environment changes.  We do not aim for a precise definition of cooperation, which is practically infeasible, but rather define necessary conditions from a human-perspective to promote cooperative behaviors.  More refined forms of cooperative rewards are discussed in **Appendix I: TASK-RELEVANT COOPERATIVE INTRINSIC REWARD (EXTENSION)**, which introduces *task-aware cooperative rewards*. This addresses Reviewer e8pD's concerns and is also highlighted in **Limitations and Future Works**.

3. **Dynamic Weighting Strategy**

   Reviewer yUUJ suggested further explanation of the dynamic weighting strategy. We have added paragraphs in the manuscript to clarify its significance. The underlying intuition is that, during the early stages of training, uncertainty-driven exploration helps agents quickly discover potentially valuable regions of the state space, whereas later in training, improving cooperative capabilities becomes more critical for overall task performance in multi-agent settings.

4. **Dual-Policy Strategy**

   Reviewer e8pD asked why the Dual-Policy Strategy is necessary. In MARL, credit assignment and non-stationarity are intrinsic challenges. Task rewards must be distributed among agents, and directly adding exploration rewards to task rewards exacerbates credit assignment difficulty and non-stationarity. Our dual-policy architecture separates the learning of exploratory action values from task-related rewards, mitigating these issues and enabling stable long-term learning of exploratory policies.

### Sufficiency of Experiments

1. **Hyperparameter tuning and future directions for adaptive exploration parameters**

   Reviewer yUUJ suggested including more discussion on hyperparameters. In **Appendix E: Additional Experimental Results**, we analyze the influence and robustness of key hyperparameters. Furthermore, in **Limitations and Future Work**, we highlight that *developing adaptive mechanisms to combine these two types of exploration rewards is an interesting avenue for future research*, which strengthens the paper.

2. **Convergence of the dynamic model, computational complexity, and the effect of ensemble size \(K\)**

   Reviewer e8pD raised concerns regarding this point. In RL and MARL, it is common to introduce additional learnable modules to generate exploration rewards, such as RND, ICM, Disagreement, and Ensembles. These modules, which supervise the environment dynamics, gradually converge as training progresses. The choice of \(K\) affects computational efficiency. In our experiments, we adopt \(K\) consistent with prior literature, providing sufficiently accurate uncertainty estimates while balancing computational cost. Although computational cost increases with the number of agents, our results show that training time remains manageable. A detailed discussion of training time for all considered algorithms is provided in **Appendix H: TRAINING TIME**. To further address the reviewer’s concern and enrich the paper, we conducted experiments with different ensemble sizes \(K = 3, 5, 8, 10\). The results, shown in **Appendix M: EFFECT OF K**, validate the reasonableness of the default choice.

3. **Sufficiency of evidence for promoting cooperative behavior**

   Reviewer yUUJ questioned whether the evidence for cooperative behavior is sufficient, noting that although the provided figures visualizing the exploration policy demonstrate emergent cooperation, this might be insufficient. In prior MARL work, emergent cooperation is commonly illustrated through policy visualization. This is because cooperation-oriented exploration rewards are heuristic and human-perspective based: they define necessary conditions for real cooperation, which naturally promotes cooperative exploration. Therefore, quantitative evaluation is rarely used. Our paper provides extensive visualization results that confirm the learned exploratory actions exhibit cooperative characteristics. We have provided a detailed response to Reviewer yUUJ addressing this point.

---

### Author Response · Authors · 2025-12-01
**Summary comment to AC (part 1/2)**

Dear AC,

We sincerely thank the reviewers for their valuable comments. Below we summarize the major revisions we made to address the reviewers' concerns and to strengthen the paper.

### Clarification on Related Works and Contributions

1. **Strengthened related works and clarified contribution boundaries.**

   Several reviewers suggested expanding the discussion of related work:

- Reviewer 6ZfR asked for clearer positioning and why our method is a meaningful step.

- Reviewer yUUJ requested discussion on ensemble value function works.

- Reviewer e8pD suggested adding more discussion on limitations of current approaches.

  To address this, we significantly expanded **Appendix L (RELATED WORKS)**, where we:

- conducted a comprehensive survey of exploration methods in MARL;
- discussed the limitations of previous methods and ensemble models in RL;
- clarified our contributions, covering the Challenge, Method, Uniqueness, Future Direction.

2. **Research gap, primary technical novelty, and core contributions**

   We clarified the research gap as follows:

   - existing MARL exploration methods struggle to generate reliable signals in high-dimensional state–action spaces;
   - prior works treat diversity-driven and cooperation-driven exploration separately, lacking a unified perspective.

   Our primary technical novelty includes:

   - introducing ensemble-based environment models into MARL exploration to address high-dimensional uncertainty estimation;
   - developing an information-theoretic cooperation reward based on ensemble dynamics, enabling ensembles to jointly capture diversity and cooperative structure;
   - designing a dynamic weighting mechanism that balances epistemic (diversity) and cooperative exploration signals during training;
   - proposing a dual-policy framework that separates exploration and main policies to reduce interference and allow the two signals to complement each other.

   Key contributions and meaningful impact

   - Our method addresses two long-standing MARL challenges: large state–action spaces and the separation between diversity-driven and cooperation-driven exploration. We present a unified, theoretically grounded framework that integrates epistemic uncertainty and cooperative influence. In other words, by leveraging **ensemble-based environment models**, our approach unifies these two major types of MARL exploration methods. Looking forward, it has the potential to further unify **goal-conditioned exploration** and achieve more realistic cooperative exploration.

3. **Reviewer 6ZfR noted similarities with previous techniques; we clarified that:**

    - Ensemble-based environment models have not been used for MARL exploration;
    - Our information-theoretic cooperation reward differs fundamentally from previous cooperative bonuses;
    - A dynamic weighting mechanism has not been previously developed.
    - Our dual-policy architecture specifically stabilizes exploration and avoids introducing additional non-stationarity and credit assignment issues. This design is not identical to prior work.
    - A paper that completely avoids using or extending any known techniques would set an unrealistically high bar. Although some underlying techniques have been explored in prior work, our method does **not** simply combine existing components. Instead, we identify core limitations in MARL exploration and propose **theoretically grounded and novel solutions that directly address these challenges**. We believe this constitutes a meaningful contribution to the design of exploration rewards and exploration strategies in MARL, and provides new insights for the community. For a detailed discussion of the differences and novelties of our approach compared with prior work, please refer to **Response to Reviewer 6ZfR** and **Appendix L (RELATED WORKS)**.

---

### Note · Authors · 2026-02-05

I have read and agree with the venue's withdrawal policy on behalf of myself and my co-authors.

---

### Meta-Review · Area_Chair_gP7X · 2026-01-04

**Summary:**

I will list the most important comments that the reviewers noted during the review process:
1) A more precise definition of the research gap authors address.
2) The paper's primary contribution appears to be the specific synthesis of several powerful techniques.
3) The authors don’t provide the proof of their choice of using the ensemble to predict state variance.
4) The authors  don't provide justification as to why dynamic weighting strategy is effective.
5) Some highly relevant works aren't discussed or compared against.
6) No information was provided regarding how hyperparameters should be tuned.
7) EECE needs a reliable model of environment dynamics to support epistemic and cooperative exploration.
8) The ablation results show limited performance improvement.

**Reviewer Concerns:**

The authors responded to most of the reviewers' comments:
1) Novelty: authors explain that effectively combining cooperation-driven and diversity-driven exploration is critical for efficient MARL.
2) Technical contributions: authors train the exploration policy solely with the combined exploration reward and employ two separate value decomposition methods.
3) Highly relevant works: authors have expanded the related work section to include additional relevant studies.
4) Hyperparameters: authors briefly analyze the influence and robustness of the hyperparameters.
5) Convergence: authors provide a detailed discussion of training time for all considered algorithms.

The authors gave only a partial answer to some of the comments:
1) State variance prediction: authors provide some clarification.
2) Role of the dynamic weighting strategy: authors provide some clarification.
3)  Dual-Policy Strategy: authors give some explanations.


The fully uncovered comments include the fact that the authors did not add additional baselines to the experimental part. Taking into account this and a number of incompletely resolved issues during the rebuttal phase, I believe that the work should be rejected.

**Reviewer Scores:**

1) Reviewer 6ZfR (score 4) could raise his score.
2) Reviewer yUUJ (score 6) would most likely have left his initial score.
3) Reviewer e8pD (score 4) could raise his score.

---

### Decision · Program_Chairs · 2026-01-26

Reject